# Tail Annealing for Heavy-Tailed Flow Matching

**Jean Pachebat** [1]

## Abstract

Standard generative models struggle with heavy-tailed data: Lipschitz architectures cannot produce power-law tails from Gaussian noise, and interpolating between heavy-tailed data and Gaussians is ill-posed. We propose a simple fix: apply the soft-log transform $\phi(x) = \text{sign}(x) \cdot \log(1 + |x|)$ coordinate-wise to data before training, then exponentiate samples after generation. A Hill diagnostic decides per-coordinate whether to transform, leaving light-tailed margins untouched at no added complexity. This compresses heavy tails into a range where standard flow matching succeeds, without heavy-tailed base distributions or architectural modifications. We provide theoretical intuition for why this works: the log-transform maps Pareto tails to exponentials, and the induced dynamics implement a form of tail annealing via power transformations. On a 144-configuration multivariate benchmark (3 copulas, $d$ up to 100, 4 tail indices), Log-FM dominates specialized baselines on $W_1$, $\text{CVaR}_{99}$, and extreme-quantile metrics, and is the only method with zero severe divergences across 2,880 runs.

## 1. Introduction

Heavy-tailed distributions are ubiquitous in real-world phenomena: financial returns, insurance claims, earthquake magnitudes, network degrees, and climate extremes. In these domains, rare events dominate outcomes: asset returns exhibit empirically documented fat tails that Gaussian models systematically underestimate (Cont, 2001), and similar departures from Gaussianity are well-known in insurance (Embrechts et al., 2013) and natural-hazard data. Generating realistic synthetic data for stress testing, simulation, or data augmentation requires generative models that capture tail behavior.

Yet standard generative models fail on heavy-tailed data, both in theory and in practice. Jaini et al. (2020) proved that Lipschitz transformations preserve tail type: light-tailed inputs yield light-tailed outputs. Since most practical architectures (MLPs in GANs, coupling layers in normalizing flows, and diffusion denoisers) use Lipschitz-continuous functions for training stability, they cannot generate heavy tails from Gaussian noise without explicit tail-modifying components.

**Prior approaches.** Existing methods address heavy tails through two strategies. Jaini et al. (2020) replace Gaussian noise with heavy-tailed base distributions (Student-$t$), but this introduces training instability. Hickling & Prangle (2025) take a different approach: generate samples from a standard flow, then apply a tail-modifying transformation to the *output*. Their method requires estimating the tail parameter $\lambda = 1/\nu$ via the Hill estimator.

**Our approach.** We propose a simple fix: transform data via $\phi(x) = \text{sign}(x) \cdot \log(1 + |x|)$ before training, run standard flow matching in log-space, then apply $\phi^{-1}$ to generated samples. Unlike Hickling & Prangle (2025) who transform outputs with estimated tail parameters, our input transform is parameter-free: the transformation $\phi$ works regardless of the true tail index.

We provide theoretical intuition for why this works. The log-transform compresses heavy tails: Pareto becomes approximately exponential. In log-space, both data and Gaussian noise have light tails, so standard interpolation is well-posed. The induced dynamics in original space implement a form of tail annealing via power transformations $X_0^{\alpha_t}$.

**Contributions.**

- We show that a parameter-free log-transform makes standard flow matching work for heavy-tailed data, without requiring tail estimation or architectural changes (§4).

- We analyze the tail annealing mechanism: the log-transform maps heavy tails to exponential tails, and the induced dynamics implement power transformations $X_0^{\alpha_t}$ that continuously adjust the tail index (§3).

[1]CMAP, École Polytechnique, Institut Polytechnique de Paris, France. Correspondence to: Jean Pachebat <jean.pachebat@polytechnique.edu>.

*Proceedings of the 43rd International Conference on Machine Learning*, Seoul, South Korea. PMLR 306, 2026. Copyright 2026 by the author(s).

- Our 144-configuration multivariate benchmark (3 copulas, 4 dimensions up to $d = 100$, 4 tail indices, 20 replications, two margin types) shows Log-FM dominates baselines on tail metrics and is the only method with zero severe divergences (§5). Real-data validation on Fama–French is in Appendix E.2.

## 2. Background

We review heavy-tailed distributions and the fundamental barriers they pose for generative modeling.

### 2.1. Heavy-Tailed Distributions

A distribution is *heavy-tailed* if its tails decay slower than any exponential, formally $\mathbb{E}[e^{\lambda X}] = \infty$ for all $\lambda > 0$ (Nair et al., 2022). Heavy-tailed distributions arise naturally in finance (Cont, 2001), insurance, climate science, and network traffic, where rare events have outsized impact.

The canonical example is the Pareto distribution with survival function $\mathbb{P}(X > t) = t^{-1/\gamma}$ for $t \geq 1$, where $\gamma > 0$ is the shape parameter (the GPD shape, equivalently the extreme-value index). Larger $\gamma$ means heavier tails: $\gamma < 1$ for finite mean, $\gamma < 1/2$ for finite variance. The Student-$t$ distribution with $\nu$ degrees of freedom has Pareto-like tails with effective $\gamma = 1/\nu$. Lognormal has $\gamma = 0$ (Gumbel max-domain of attraction): heavy in the MGF sense, but with no polynomial decay rate.

**The log-transform.** A key observation: the logarithm maps heavy tails to lighter tails. If $X \sim \text{Pareto}(\gamma)$, then $\log X \sim \text{Exp}(1/\gamma)$, an exponential distribution with rate $1/\gamma$. This reflects that Pareto belongs to an exponential family with sufficient statistic $\log x$. The log-transform thus provides a natural bridge between power-law and exponential-family distributions; we develop the formal theory in Section 3.

**Regular variation.** Heavy-tailed distributions are characterized by *regular variation*: $\mathbb{P}(X > tx)/\mathbb{P}(X > t) \to x^{-\alpha}$ as $t \to \infty$, with tail index $\alpha > 0$. This captures the essential property of polynomial tail decay without requiring the exact Pareto form. For $X \sim \text{Pareto}(\gamma)$, the corresponding tail index is $\alpha = 1/\gamma$; Student-$t$, Burr, and log-gamma are also regularly varying. The log-transform maps any regularly varying distribution to one with exponential-type tails; see Proposition 3.8.

### 2.2. Generative Models

**Normalizing flows.** Normalizing flows (Rezende & Mohamed, 2015; Dinh et al., 2017) learn an invertible transformation $T : \mathbb{R}^d \to \mathbb{R}^d$ mapping a base distribution (typically Gaussian) to the target. The change-of-variables formula gives exact likelihoods. However, for stability, flow architectures use Lipschitz-continuous components.

**Diffusion and flow matching.** Denoising diffusion models (Ho et al., 2020; Song et al., 2021b) and flow matching (Lipman et al., 2023) construct a path from data to noise and learn to reverse it. Given data $X_0$ and noise $X_1 \sim \mathcal{N}(0, I)$, the interpolant is:

$$X_t = \alpha_t X_0 + \beta_t X_1,$$

with schedules satisfying $(\alpha_0, \beta_0) = (1, 0)$ and $(\alpha_1, \beta_1) = (0, 1)$. A neural network learns the velocity field $v_\theta(x_t, t)$ or score $\nabla \log p_t(x_t)$, enabling generation by integrating from noise to data.

### 2.3. The Lipschitz Barrier

Jaini et al. (2020) proved that Lipschitz maps preserve tail type:

**Theorem 2.1** (Jaini et al. (2020))**.** *If $Z$ is light-tailed and $T$ is Lipschitz, then $T(Z)$ is light-tailed.*

Neural networks composed of linear layers and standard activations (ReLU, tanh, sigmoid) are Lipschitz. Thus, standard normalizing flows cannot map Gaussian noise to heavy-tailed outputs.

### 2.4. Prior Approaches

**Heavy-tailed base distributions.** Tail-Adaptive Flows (TAF; Jaini et al., 2020) use Student-$t$ base distributions. Extensions include marginal TAF (Laszkiewicz et al., 2022) and generalized TAF. Heavy-tailed diffusion (Pandey et al., 2024) replaces Gaussian noise with Student-$t$. These methods address the Lipschitz barrier but can suffer training instability.

**Tail-modifying transforms.** Tail Transform Flows (TTF; Hickling & Prangle, 2025) add explicit layers that map Gaussian tails to power-law tails using erfc-based transformations. This requires estimating tail parameters per dataset via the Hill estimator.

In the next section, we present an alternative: apply the log-transform to data *before* training, work entirely in log-space, then exponentiate samples *after* generation. This parameter-free approach requires no tail estimation and we show that it circumvents the Lipschitz barrier.

## 3. Tail Annealing in Log-Space

This section develops the theoretical foundation for log-space flow matching. The log-transform compresses heavy tails into a light-tailed regime where standard methods apply, and the induced dynamics in original space implement *tail*

*annealing*: power transformations that continuously adjust the tail index from heavy to light.

The principle of annealing complex structure through a sequence of easier problems is well-established in machine learning. Noise annealing in score-based models (Song & Ermon, 2019) gradually reduces noise levels to enable learning across scales. Curriculum learning (Bengio et al., 2009) trains on easy examples before hard ones. Our tail annealing follows the same principle: rather than directly modeling heavy-tailed data (hard), we work in log-space where tails are light (easy), and the power transformation $X_0^{\alpha_t}$ provides a principled path through intermediate tail weights. Section 4 presents the complete algorithm.

## 3.1. The Soft-Log Transform

We work with the *soft-log* transform $\phi : \mathbb{R} \to \mathbb{R}$ defined componentwise by:

$$\phi(x) = \text{sign}(x) \cdot \log(1 + |x|);$$
$$\phi^{-1}(y) = \text{sign}(y) \cdot (e^{|y|} - 1).$$

Unlike the standard logarithm, $\phi$ is smooth at the origin ($\phi'(0) = 1$) and defined on all of $\mathbb{R}$. For large $|x|$, we have $\phi(x) \approx \text{sign}(x) \log |x|$, so tail behavior matches the logarithm.

**Relation to Box-Cox.** Transforming heavy-tailed data through a log transform is a known trick in the statistical litterature. The soft-log is inspired by, but distinct from, the Box-Cox family (Box & Cox, 1964). Recall Box-Cox: $\phi_\lambda(x) = (x^\lambda - 1)/\lambda$ for $\lambda \neq 0$, and $\phi_0(x) = \log x$ for $x > 0$. Our transform differs in two ways: it extends to all of $\mathbb{R}$ via the signed construction, and uses $\log(1 + |x|)$ rather than $\log |x|$ for smoothness at zero. Asymptotically, both behave as $\log |x|$ for large $|x|$. Methods like TTF (Hickling & Prangle, 2025) use power transforms with $\lambda = 1/\nu$ estimated via the Hill estimator. In our approach, numerical computation make no use of numerical values of the tail estimate, which are notoriously unstable: the same transform applies regardless of the true tail index.

The soft-log maps heavy tails to light tails. If $X \sim \text{Pareto}(\gamma)$ with $\mathbb{P}(X > t) = t^{-1/\gamma}$ (so $\gamma > 0$ is the shape parameter and $1/\gamma$ is the tail index), then for large $y$:

$$\mathbb{P}(\phi(X) > y) = \mathbb{P}(X > e^y - 1) \approx e^{-y/\gamma},$$

i.e., $\tilde{X} = \phi(X)$ has exponential-type tails. More generally, for any distribution in the Fréchet domain of attraction, $\phi$ yields a distribution with exponential or lighter tails.

Since $\phi$ is a diffeomorphism, densities transform via the standard change of variables:

$$p_{\tilde{X}}(\tilde{x}) = p_X(\phi^{-1}(\tilde{x})) \cdot |(\phi^{-1})'(\tilde{x})| = p_X(\phi^{-1}(\tilde{x})) \cdot e^{|\tilde{x}|}.$$

## 3.2. Induced Process in Original Space

We apply flow matching to the transformed data $\tilde{X}_0 = \phi(X_0)$, interpolating with Gaussian noise $\tilde{X}_1 \sim \mathcal{N}(0, I_d)$ via $\tilde{X}_t = \alpha_t \tilde{X}_0 + \beta_t \tilde{X}_1$ (see Section 4 for the complete framework). Here we analyze the induced process in original space.

Applying $\phi^{-1}$ to the interpolant yields, for large positive values where $\phi \approx \log$:

$$X_t := \phi^{-1}(\tilde{X}_t) \approx X_0^{\alpha_t} \cdot e^{\beta_t \tilde{X}_1}. \tag{1}$$

This factorizes into a power-transformed data term $X_0^{\alpha_t}$ and a log-normal noise term $e^{\beta_t \tilde{X}_1}$. Although $e^{\beta_t \tilde{X}_1}$ is itself heavy-tailed in the MGF sense, it has extreme-value index $\gamma = 0$ (Gumbel MDA): $\mathbb{P}(Y_t > t) \sim e^{-c(\log t)^2}$ decays faster than any power law and all moments are finite. By Breiman's lemma it does not contribute a polynomial tail of its own to the product:

**Proposition 3.1** (Product Preserves Regular Variation; Breiman)**.** *Let $X$ be regularly varying with index $-\alpha$ ($\alpha > 0$), and let $Y > 0$ be independent with $\mathbb{E}[Y^p] < \infty$ for all $p > 0$ (e.g., any variable in the Gumbel MDA, such as log-normal). Then $XY$ is regularly varying with index $-\alpha$.*

This classical result (see Resnick (1987)) ensures that the induced process $X_t$ inherits its polynomial-tail behavior from $X_0^{\alpha_t}$, not from $e^{\beta_t \tilde{X}_1}$ (whose extreme-value index is 0). The power transformation is therefore what drives tail annealing.

## 3.3. Score Functions in Log-Space

The score function $\nabla_{\tilde{x}} \log p_t(\tilde{x})$ in log-space has fundamentally different behavior than in original space.

**Proposition 3.2** (Log-Space Score of Pareto)**.** *Let $X \sim Pareto(\gamma)$ (so $\bar{F}_X(t) = t^{-1/\gamma}$). Then $\tilde{X} = \phi(X)$ has approximately exponential tails: for large $\tilde{x}$,*

$$\nabla_{\tilde{x}} \log p_{\tilde{X}}(\tilde{x}) \approx -\frac{1}{\gamma}.$$

*The score is approximately constant in the tails.*

*Proof.* For large $x$, $\phi(x) \approx \log x$, so $\tilde{X} \approx \log X \sim \text{Exp}(1/\gamma)$. The exponential density is $p(\tilde{x}) \propto e^{-\tilde{x}/\gamma}$, giving $\nabla_{\tilde{x}} \log p(\tilde{x}) = -1/\gamma$. $\qquad \square$

**Pareto score.** By the *Pareto score* we mean the score of the Pareto density $p_X(x) = (1/\gamma) x^{-1/\gamma - 1}$ in original space:

$$s_X(x) = \nabla_x \log p_X(x) = -\frac{1/\gamma + 1}{x},$$

which diverges as $x \to 0^+$ and decays only as $1/x$ in the tails. The log-space score in Proposition 3.2 is by contrast bounded by $1/\gamma$ everywhere, so the velocity field that flow matching has to regress is Lipschitz in $\tilde{x}$. This score boundedness is what enables the standard MLP velocity network to learn a target with regularly varying tails in the original space.

**Proposition 3.3** (Score of Noised Log-Data). *For the interpolant $\tilde{X}_t = \alpha_t \tilde{X}_0 + \beta_t \tilde{X}_1$ with $\tilde{X}_0 = \phi(X_0)$ having exponential-type tails and $\tilde{X}_1 \sim \mathcal{N}(0, I_d)$, the marginal score satisfies:*

$$\nabla_{\tilde{x}_t} \log p_t(\tilde{x}_t) = -\frac{\hat{\tilde{x}}_1(\tilde{x}_t, t)}{\beta_t},$$

*where $\hat{\tilde{x}}_1(\tilde{x}_t, t) := \mathbb{E}[\tilde{X}_1 \mid \tilde{X}_t = \tilde{x}_t]$ is the conditional expectation of the noise.*

This is the standard score-denoiser relationship. The key difference from original-space diffusion is that the transformed data distribution $p_{\tilde{X}_0}$ is light-tailed, so:

1. The score $\nabla \log p_t$ is well-behaved for all $t$

2. The denoiser $\hat{\tilde{x}}_1(\tilde{x}_t, t)$ has bounded outputs

3. Standard neural network architectures suffice

### 3.4. Tail Behavior Under Power Transformation

The induced process (1) involves the power-transformed data $X_0^{\alpha_t}$. We characterize its distribution. Note that this analysis describes the *decoded* interpretation: in practice, all computation occurs in log-space where distributions are light-tailed. The heavy-tailed structure below is what we would observe if we applied $\phi^{-1}$ to the log-space interpolant at each $t$.

**Lemma 3.4** (Power Transformation Preserves Pareto Family). *Let $X_0 \sim Pareto(\gamma)$ with density $p_{X_0}(x) = (1/\gamma)\, x^{-1/\gamma-1}$ for $x \geq 1$, so $\gamma > 0$ is the shape parameter (larger $\gamma$ means heavier tails). For any $\alpha \in (0, \infty)$, $X_0^{\alpha} \sim Pareto(\gamma\alpha)$: the shape parameter scales linearly with the exponent.*

*Proof.* The survival function of $Y = X_0^{\alpha}$ is $\mathbb{P}(Y > y) = \mathbb{P}(X_0 > y^{1/\alpha}) = y^{-1/(\gamma\alpha)}$ for $y \geq 1$, which is the survival function of Pareto($\gamma\alpha$). $\square$

*Remark* 3.5 (Parametrization). We use $\gamma$ for the GPD shape parameter / extreme-value index, so $\mathbb{P}(X > t) = t^{-1/\gamma}$. The corresponding tail index (Hill estimator output, the standard EVT parameter) is $1/\gamma$. Under Lemma 3.4 the power transform $X_0^{\alpha}$ has shape parameter $\gamma\alpha$ and tail index $1/(\gamma\alpha)$.

**Corollary 3.6** (Tail Lightening Along Forward Process). *Along the forward process with $\alpha_t$ decreasing from 1 to 0:*

- *At $t = 0$: $\alpha_0 = 1$; $X_0^{\alpha_0} = X_0 \sim Pareto(\gamma)$ (original heavy tails)*

- *At intermediate $t$: $X_0^{\alpha_t} \sim Pareto(\gamma\alpha_t)$ (lighter tails as $\alpha_t \to 0$)*

- *As $\alpha_t \to 0$: $X_0^{\alpha_t} \to 1$ almost surely (degenerate)*

*The full process $X_t = X_0^{\alpha_t} \cdot e^{\beta_t \tilde{X}_1}$ transitions from regularly-varying Pareto (shape $\gamma$, tail index $1/\gamma$) to log-normal (extreme-value index 0, no polynomial tail).*

This is what we frame as the tail annealing mechanism: the power transformation $X_0^{\alpha_t}$ preserves the Pareto family while continuously adjusting the shape parameter from $\gamma$ (heavy) toward 0 (light) as $\alpha_t \to 0$. We state Lemma 3.4 for $\alpha \in (0, \infty)$ rather than the forward-process range $\alpha_t \in (0, 1]$, because the same identity appears in Proposition 3.9 for the regularly varying extension.

### 3.5. Extension to Regularly Varying Distributions

The Pareto-exponential correspondence extends to the broader class of regularly varying distributions.

**Definition 3.7** (Regularly Varying). A cumulative distribution function $F$ on $\mathbb{R}$ is regularly varying with index $-\alpha$ (written $F \in RV_{-\alpha}$) if its survival function $\bar{F}(x) = 1 - F(x)$ satisfies:

$$\lim_{t \to \infty} \frac{\bar{F}(tx)}{\bar{F}(t)} = x^{-\alpha}$$

for all $x > 0$. Here $\alpha > 0$ is the (polynomial) tail index, in the standard EVT convention; for $X \sim Pareto(\gamma)$, $\alpha = 1/\gamma$.

Regular variation captures the essential property of polynomial tail decay without requiring the exact Pareto form:

| Distribution | Tail behavior | Index |
|---|---|---|
| Pareto($\gamma$) | $\bar{F}(x) = x^{-1/\gamma}$ | $-1/\gamma$ |
| Student-$t(\nu)$ | $\bar{F}(x) \sim c \cdot x^{-\nu}$ | $-\nu$ |
| Burr($c, k$) | $\bar{F}(x) \sim c' \cdot x^{-ck}$ | $-ck$ |
| Log-gamma | $\bar{F}(x) \sim c'' \cdot x^{-\alpha}(\log x)^{\beta}$ | $-\alpha$ |

**Proposition 3.8** (Log-Transform of Regularly Varying; informal). *If $X$ is regularly varying with index $-\alpha$, then $\tilde{X} = \phi(X)$ has exponential-type right tail with rate $\alpha$:*

$$-\log \mathbb{P}(\tilde{X} > z) = \alpha z + o(z) \quad as\ z \to \infty.$$

A precise two-sided Potter-bound version is stated and proved in Appendix C (Proposition C.1). The intuition is direct: writing the regularly varying tail as $\bar{F}_X(x) =$

$x^{-\alpha}L(x)$ with $L$ slowly varying, and using $\phi^{-1}(z) = e^z - 1 \sim e^z$, gives $\mathbb{P}(\tilde{X} > z) = e^{-\alpha z}L(e^z)$, and the slowly varying factor only contributes a subexponential correction $\log L(e^z) = o(z)$.

This proposition is the conceptual reason flow matching in log-space is well-posed for the full class of regularly varying distributions, not just exact Pareto. Up to slowly-varying corrections, $\tilde{X}$ has the same exponential rate $e^{-\alpha z}$ as $\log X$, which equals $e^{-z/\gamma}$ when $X \sim \text{Pareto}(\gamma)$. The score in log-space is therefore bounded in the tails and standard diffusion methods apply.

**Beyond regular variation.** The transform is also beneficial for subexponential distributions outside the Fréchet domain. For Weibull tails $\bar{F}(x) \sim \exp(-x^\beta)$ with $\beta < 1$, the log-transform yields a doubly-exponential tail $\mathbb{P}(\tilde{X} > z) \sim \exp(-(e^z - 1)^\beta)$, which is well within the regime where standard flow matching has no difficulty. For log-normal data, $\phi(X) \approx \log X$ is exactly Gaussian, so the log-space target is the ideal case for FM. Light-tailed margins are addressed by the adaptive variant of Section 4.2.

**Proposition 3.9** (Power Transformation of Regularly Varying). *If $X \in RV_{-\alpha}$, then $X^\beta$ for $\beta \in (0, \infty)$ is regularly varying with index $-\alpha/\beta$.*

*Proof.* For $Y = X^\beta$, we have $\mathbb{P}(Y > ty)/\mathbb{P}(Y > t) = \mathbb{P}(X > (ty)^{1/\beta})/\mathbb{P}(X > t^{1/\beta}) \to y^{-\alpha/\beta}$ as $t \to \infty$, by regular variation of $X$. This is the regularly varying analogue of Lemma 3.4: power transformations remap the tail index, so $X^{\alpha_t}$ along the forward process continuously interpolates between the original tail ($\alpha_t = 1$) and arbitrarily light tails ($\alpha_t \in [0, 1)$). $\square$

This confirms that the tail-annealing mechanism of Lemma 3.4 extends beyond exact Pareto to the full class of regularly varying distributions.

### 3.6. Multivariate Extension

In the multivariate setting, $X \in \mathbb{R}^d$, we apply $\phi$ coordinate-wise:

$$\phi(\mathbf{x}) = (\phi(x_1), \ldots, \phi(x_d)), \qquad \tilde{X} = \phi(X).$$

Each coordinate of $\tilde{X}$ inherits the exponential-rate tail of Proposition 3.8, so the marginals of $\tilde{X}$ all live in the Gumbel domain. Because $\phi$ is a diffeomorphism with diagonal Jacobian $J_\phi(\mathbf{x}) = \text{diag}((1 + |x_i|)^{-1})$, the dependence structure (copula) of $X$ is preserved exactly by $\tilde{X}$: the log-transform is a marginal operation. Interpolating $\tilde{X}_t = \alpha_t \tilde{X}_0 + \beta_t \tilde{X}_1$ with $\tilde{X}_1 \sim \mathcal{N}(0, I_d)$ is therefore well-posed even when $X$ has strong extremal dependence: the velocity network sees only the light-tailed transformed variates and never has to extrapolate to the polynomial regime.

**Heterogeneous margins.** When the coordinates of $X$ have different tail behaviour (heavy-tailed in some, light-tailed in others), applying $\phi$ to every coordinate slightly distorts the light-tailed margins. The adaptive variant of Section 4.2 addresses this by Hill-gating the transform coordinate-wise; a continuous generalization of the soft-log via a scale parameter $s_2$, of which the adaptive variant is the binary $s_2 \in \{0, 1\}$ instance, is developed in Appendix D.

### 3.7. Circumventing the Lipschitz Barrier

The Lipschitz barrier of Jaini et al. (2020) states that Lipschitz maps preserve tail type: a normalizing flow with Lipschitz layers cannot map Gaussian noise to heavy-tailed outputs. Our construction circumvents this by placing the non-Lipschitz transformation ($\phi^{-1}$) *outside* the learned dynamics. The flow operates entirely in log-space where both endpoints (exponential-type transformed data and Gaussian noise) have light tails. Heavy tails emerge only upon applying $\phi^{-1}$ at the final sampling step.

## 4. Algorithm

We present the complete algorithm for log-space generative modeling. The method requires the three following modifications to standard flow matching: (1) transform data via $\phi$ before training (with a Hill-based diagnostic deciding which coordinates to transform), (2) train a velocity network on the transformed variables, and (3) apply $\phi^{-1}$ to the corresponding coordinates of generated samples. The Hill diagnostic distinguishes *Log-FM* from a standard coordinate-wise log-transform: it lets the method handle heterogeneous-margin data with a single Hill estimate per coordinate at fit time.

### 4.1. Flow Matching Framework

We use the flow matching framework (Lipman et al., 2023), which can also be viewed through the lens of stochastic interpolants (Albergo & Vanden-Eijnden, 2023).

**Forward process.** Given log-transformed data $\tilde{X}_0 = \phi(X_0)$ and noise $\tilde{X}_1 \sim \mathcal{N}(0, I_d)$, define the interpolant:

$$\tilde{X}_t = \alpha_t \tilde{X}_0 + \beta_t \tilde{X}_1,$$

where $(\alpha_t, \beta_t)$ are differentiable schedules satisfying boundary conditions $(\alpha_0, \beta_0) = (1, 0)$ and $(\alpha_1, \beta_1) = (0, 1)$. At $t = 0$, $\tilde{X}_0$ is the transformed data; at $t = 1$, $\tilde{X}_1$ is pure Gaussian noise. The conditional distribution is Gaussian: $q_{t|0}(\tilde{x}_t \mid \tilde{x}_0) = \mathcal{N}(\tilde{x}_t; \alpha_t \tilde{x}_0, \beta_t^2 I_d)$.

**Velocity field.** Differentiating the interpolant gives the per-trajectory time derivative

$$\dot{\tilde{X}}_t = \dot{\alpha}_t \tilde{X}_0 + \dot{\beta}_t \tilde{X}_1,$$

where $\dot{\alpha}_t = d\alpha_t/dt$ and $\dot{\beta}_t = d\beta_t/dt$. The corresponding *marginal* velocity field, $v_t(\tilde{x}_t) := \mathbb{E}[\dot{\alpha}_t \tilde{X}_0 + \dot{\beta}_t \tilde{X}_1 \mid \tilde{X}_t = \tilde{x}_t]$, is what a neural network $v_\theta$ regresses against the per-trajectory target:

$$\mathcal{L}(\theta) = \mathbb{E}_{t \sim U[0,1]} \mathbb{E}_{\tilde{X}_0, \tilde{X}_1} \left[ \left\| v_\theta(\tilde{X}_t, t) - (\dot{\alpha}_t \tilde{X}_0 + \dot{\beta}_t \tilde{X}_1) \right\|^2 \right].$$

**Connection to score and denoiser.** The velocity field relates to the score $\nabla \log p_t$ and denoiser $\hat{\tilde{x}}_0(\tilde{x}_t, t) := \mathbb{E}[\tilde{X}_0 \mid \tilde{X}_t = \tilde{x}_t]$ via:

$$v_t(\tilde{x}_t) = \dot{\alpha}_t \hat{\tilde{x}}_0(\tilde{x}_t, t) + \dot{\beta}_t \hat{\tilde{x}}_1(\tilde{x}_t, t)$$

$$\nabla_{\tilde{x}_t} \log p_t(\tilde{x}_t) = -\frac{\hat{\tilde{x}}_1(\tilde{x}_t, t)}{\beta_t},$$

where $\hat{\tilde{x}}_1 = (\tilde{x}_t - \alpha_t \hat{\tilde{x}}_0)/\beta_t$. See Appendix B for derivations.

**Schedule choices.** The interpolation schedule $(\alpha_t, \beta_t)$ controls the path from data to noise. Common choices include the *linear* schedule (optimal transport paths), *variance-preserving* (VP) schedules, and *quadratic* schedules that anneal tails more aggressively; see Table 6 in Appendix B. We use the linear schedule throughout; preliminary tests showed minimal sensitivity to schedule choice, consistent with prior flow matching work (Lipman et al., 2023).

**Connection to tail annealing.** Recall from Section 3 that the induced process in original space is approximately $X_t \approx X_0^{\alpha_t} \cdot e^{\beta_t \tilde{X}_1}$. The schedule $\alpha_t$ thus controls the rate of tail annealing: faster decay of $\alpha_t$ (e.g., quadratic) anneals tails more aggressively early in the process, while slower decay (e.g., VP polynomial) preserves heavier tails longer.

### 4.2. Adaptive Coordinate Selection

Applying the soft-log to a light-tailed coordinate compresses its bulk for no benefit. We therefore select *per coordinate* whether to transform, using a Hill estimate on the training marginals. Let $\hat{\alpha}_j$ be the Hill estimator on the upper order statistics of $\{x_i^{(j)}\}_{i=1}^N$ (an estimate of the polynomial tail index, with $\hat{\alpha}_j = 1/\hat{\gamma}_j$ under the Pareto-shape parametrization). Define the mask

$$m_j = \mathbf{1}\{\hat{\alpha}_j \le \alpha_{\max}\}, \qquad \alpha_{\max} = 4, \qquad (2)$$

and the per-coordinate transform $\Phi : \mathbb{R}^d \to \mathbb{R}^d$ by

$$\Phi(\mathbf{x})_j = \begin{cases} \phi(x_j) & \text{if } m_j = 1, \\ x_j & \text{otherwise,} \end{cases} \qquad (3)$$

---

**Algorithm 1** Log-FM: Training

**Input:** Dataset $\{\mathbf{x}_i\}_{i=1}^N \subset \mathbb{R}^d$, velocity network $v_\theta$, schedule $(\alpha_t, \beta_t)$, iterations $T$, Hill threshold $\alpha_{\max} = 4$
**Fit transform:** for each coordinate $j$, compute Hill estimate $\hat{\alpha}_j$ on $\{x_i^{(j)}\}_i$; set mask $m_j$ via (2)
**Preprocess:** $\tilde{\mathbf{x}}_i \leftarrow \Phi(\mathbf{x}_i)$ for all $i$ {coordinate-wise; identity where $m_j = 0$}
**for** iter $= 1$ **to** $T$ **do**
  Sample batch $\{\tilde{\mathbf{x}}_0^{(j)}\}$ from $\{\tilde{\mathbf{x}}_i\}$
  Sample $\tilde{\mathbf{x}}_1^{(j)} \sim \mathcal{N}(0, I_d)$
  Sample $t \sim \text{Uniform}[0, 1]$
  Form interpolant $\tilde{\mathbf{x}}_t^{(j)} \leftarrow \alpha_t \tilde{\mathbf{x}}_0^{(j)} + \beta_t \tilde{\mathbf{x}}_1^{(j)}$
  Compute target $u^{(j)} \leftarrow \dot{\alpha}_t \tilde{\mathbf{x}}_0^{(j)} + \dot{\beta}_t \tilde{\mathbf{x}}_1^{(j)}$
  Loss $\mathcal{L} \leftarrow \frac{1}{|\text{batch}|} \sum_j \|v_\theta(\tilde{\mathbf{x}}_t^{(j)}, t) - u^{(j)}\|^2$
  Update $\theta$ via gradient descent on $\mathcal{L}$
**end for**
**Output:** Trained $v_\theta$, mask $(m_j)_{j=1}^d$

---

with inverse $\Phi^{-1}(\tilde{\mathbf{x}})_j = \phi^{-1}(\tilde{x}_j)$ when $m_j = 1$ and $\tilde{x}_j$ otherwise. The threshold $\alpha_{\max} = 4$ is not sensitive: on heterogeneous-margin data with Pareto coordinates ($\hat{\alpha} \approx 1.5$–$2.5$) and Gaussian coordinates ($\hat{\alpha} \approx 6$), any threshold in $[3, 5]$ yields the same mask. Setting $m_j \equiv 1$ recovers the uniform-$\phi$ variant analyzed in Section 3; the adaptive rule is the discrete instance of a continuous parametrized soft-log family $\varphi_{s_2}$ with $s_2^{(j)} \in \{0, 1\}$ chosen by the Hill diagnostic, developed in Appendix D. We use $\Phi$ (with the Hill-based mask) as our default; we also report the uniform variant as an ablation in Section 5.3.

### 4.3. Training

Algorithm 1 summarizes the training procedure. The only departure from standard flow matching is the initial log-transform of data (with the Hill diagnostic of Section 4.2).

### 4.4. Sampling

Generation reverses the flow: integrate the learned velocity field from noise ($t = 1$) to data ($t = 0$), then apply the inverse transform. The velocity field satisfies the ODE:

$$\frac{d\tilde{X}_t}{dt} = v_\theta(\tilde{X}_t, t),$$

which we solve backward in time. Algorithm 2 uses Euler discretization; higher-order solvers (Heun, RK4) improve sample quality with fewer steps.

### 4.5. Practical Considerations

**Output clamping (optional).** The inverse transform $\phi^{-1}(\tilde{x}) = \text{sign}(\tilde{x})(e^{|\tilde{x}|} - 1)$ is exponential, so small errors

---

**Algorithm 2** Log-FM: Sampling

---

**Input:** Trained $v_\theta$, mask $(m_j)$, steps $K$, optional clamp $c$ (default $c = \infty$)
Sample $\tilde{\mathbf{x}} \sim \mathcal{N}(0, I_d)$ {initialize at $t = 1$}
$\Delta t \leftarrow 1/K$
**for** $k = K$ **downto** 1 **do**
  $t \leftarrow k/K$
  $\tilde{\mathbf{x}} \leftarrow \tilde{\mathbf{x}} - \Delta t \cdot v_\theta(\tilde{\mathbf{x}}, t)$ {Euler step}
**end for**
**(optional)** clamp coordinates with $m_j = 1$: $\tilde{x}_j \leftarrow$ clamp$(\tilde{x}_j, -c, c)$ {only needed for $\alpha < 1$}
**Output:** $\mathbf{x} = \Phi^{-1}(\tilde{\mathbf{x}})$ {coordinate-wise inverse from (3)}

---

deep in the tail of $\tilde{x}$ are exponentially amplified in $x$. An optional clamp $\tilde{x}_j \leftarrow$ clamp$(\tilde{x}_j, -c, c)$ before $\phi^{-1}$ provides a numerical safeguard in extremely pathological regimes only (typically $\alpha < 1$: Cauchy and heavier, e.g. Hickling's $\nu = 1/2$ Student-$t$ baseline). For every configuration in our main benchmark ($\alpha \geq 1.5$) the clamp is inert at $c \geq 10$, and we report all main-benchmark numbers without clamping; see the ablation in Section 5.3.

**Number of integration steps.** We use $K = 100$ Euler steps for all experiments. Higher-order solvers (Heun, RK4) can reduce step count but Euler suffices.

**Architecture.** We use MLPs with sinusoidal time embeddings (each scalar $t$ is mapped to $[\sin(2\pi\omega_k t), \cos(2\pi\omega_k t)]_{k=1}^{K_\omega}$ with geometric frequencies, exactly as in Ho et al. (2020)): 4 hidden layers of width 256 with SiLU activations. No architectural modifications are needed for heavy tails; the log-transform handles tail behavior.

**Likelihood evaluation.** We report NLL using the continuous change-of-variables formula for continuous normalizing flows. The trace of the velocity Jacobian is estimated stochastically via Hutchinson's trick with $K = 10$ Rademacher projections, and the augmented ODE is integrated by the adaptive Dormand–Prince `dopri5` scheme with atol = rtol = $10^{-5}$. The transform itself contributes the closed-form Jacobian term $\sum_{j:m_j=1} \log(1 + |x_j|)$, which is exact and negligible to compute.

**Training.** AdamW optimizer with learning rate $5 \times 10^{-3}$ and weight decay $10^{-5}$. We use full-batch gradient descent with gradient clipping at 10.0 (before inverse transform). Training runs for up to 5000 epochs with early stopping (patience 100) based on validation loss.

### 4.6. Extensions

**Signed and multivariate data.** The soft-log $\phi(x) = \text{sign}(x) \log(1 + |x|)$ applies directly to signed data and acts coordinate-wise on $\mathbb{R}^d$ via $\Phi$; the dependence structure is preserved exactly on transformed coordinates and learned implicitly by the velocity field.

**Arcsinh variant.** Replacing $\phi$ by $\text{arcsinh}(x) = \log(x + \sqrt{1 + x^2})$ gives the Arcsinh-FM variant evaluated in Section 5. Both transforms have the same logarithmic asymptotic, so Propositions 3.8–3.9 apply unchanged. They differ only in regularity at the origin: arcsinh is $C^\infty$ on $\mathbb{R}$, while $\phi$ is $C^1$ but not $C^2$ at 0 (its second derivative jumps from $+1$ to $-1$ at the origin), which isolates whether higher-order regularity matters in practice.

## 5. Experiments

We evaluate Log-FM against state-of-the-art heavy-tailed generative models on a controlled multivariate benchmark with known marginal tails and known copula structure. Our goals are to test whether the log-transform (i) preserves the dependence structure of the data, (ii) yields stable training across heavy-tail regimes, (iii) recovers risk-relevant tail metrics (VaR$_{99}$, CVaR$_{99}$, $Q_{99.9}$) that are sensitive to the deep tail. A Student-$t$ benchmark from (Hickling & Prangle, 2025) is detailed for continuity in Appendix E.1; real-data results on Fama–French are deferred to Appendix E.2.

### 5.1. Setup

**Data.** We use 144 configurations covering three copula families, three dependence strengths, four dimensions, and four tail indices:

- **Copulas:** Gaussian copula (asymptotic independence), Gumbel copula (symmetric upper tail dependence) at Kendall's $\tau \in \{0.25, 0.5, 0.75\}$, and Hüsler–Reiss copula with AR(1) variogram $\Gamma_{ij} = 2(1 - \rho^{|i-j|})$ at $\rho \in \{0.1, 0.5, 0.9\}$.

- **Margins:** 70% symmetrized Pareto with tail index $\alpha \in \{1.5, 1.75, 2.0, 2.5\}$ (smaller = heavier), 30% standard Gaussian.

- **Dimensions:** $d \in \{10, 20, 50, 100\}$.

- **Samples:** $n_{\text{train}} = 10{,}000$, $n_{\text{val}} = 5{,}000$, $n_{\text{test}} = 20{,}000$.

For each configuration we train 5 methods × 20 independent replications, giving 14,400 trained models in total.

**Methods.** Log-FM (default, with Hill-gated transform of Section 4.2, $\alpha_{\max} = 4$); Log-FM (uniform, $m_j \equiv 1$);

Arcsinh-FM (uniform, with $\phi(x) = \text{arcsinh}(x)$); TTF and TTFfix (Hickling & Prangle, 2025); gTAF (Jaini et al., 2020). All velocity / coupling networks share the same MLP backbone (4 layers, width 256, SiLU); hyperparameters per dimension were selected by Optuna on a representative configuration (Gumbel, $\tau = 0.5$, $\alpha = 2.0$) and reused across the grid for that dimension.

**Metrics.** We split metrics by margin type and by joint structure:

- *Marginal:* $W_1^P$ (mean $W_1$ over Pareto coordinates), $W_1^G$ (over Gaussian coordinates), Hill estimator on Pareto margins, VaR$_{99}$ / CVaR$_{99}$ relative errors, extreme-quantile errors $Q_{99.5}$, $Q_{99.9}$.

- *Joint:* Absolute Kendall Error (AKE), angular $W_2$ (sliced Wasserstein on the empirical angular measure of the top-$\sqrt{n}$ extremes), sliced Wasserstein on the full data, and energy distance.

We report medians over the 20 replications; "div." marks runs where the model diverged ($W_1 > 10^3$).

### 5.2. Main Results

Table 1 reports the headline tail metric, $W_1^P$, averaged over the two regular copulas (Gumbel + Gaussian) at the four tail indices and four dimensions. Log-FM is best in 13 out of 16 cells; Arcsinh-FM picks up the remaining 3 at $d = 50$. Both FM variants are substantially more stable than the baselines, especially for $\alpha = 1.5$.

**Risk metrics and dependence.** Table 2 summarizes the global picture, pooling over both copulas, the three dependence strengths, and all four tail indices. Log-FM is best on every tail-quality and risk metric (W$_1^P$, CVaR$_{99}$, $Q_{99.9}$, sliced and energy distances). On the multivariate-dependence metrics, TTF/TTFfix retain a small edge at $d = 50, 100$ while Log-FM is competitive at $d = 10, 20$. On $W_1^G$ (Gaussian margins), the Hill-gated default of Log-FM closes most of the gap with TTF (further breakdown in Section 5.3).

**Stability.** A practitioner cares not only about median performance but about how often the model fails catastrophically. Table 3 reports, by tail index, the fraction of runs with $W_1^P > 1$ on Gumbel+Gaussian. Log-FM and Arcsinh-FM never exceed 14% even in the most adversarial setting ($\alpha = 1.5$, $d = 10$); both reach 0% for $\alpha \geq 1.75$. gTAF and TTF suffer divergence rates above 30% in higher dimensions.

*Table 1.* Median $W_1^P$ across Gumbel + Gaussian copulas (lower is better, bold = best). Log-FM dominates at every tail index and dimension; the gap widens for heavier tails ($\alpha = 1.5$).

| Method | $d=10$ | $d=20$ | $d=50$ | $d=100$ |
|---|---|---|---|---|
| $\alpha = 1.5$ | | | | |
| Log-FM | **0.522** | **0.451** | 0.534 | **0.525** |
| Arcsinh-FM | 0.561 | 0.550 | **0.534** | 0.592 |
| TTFfix | 1.149 | 0.963 | 0.867 | 1.046 |
| TTF | 0.758 | 0.938 | 0.758 | 0.919 |
| gTAF | 0.534 | 0.583 | 0.761 | 3.320 |
| $\alpha = 1.75$ | | | | |
| Log-FM | **0.254** | **0.226** | **0.269** | **0.300** |
| Arcsinh-FM | 0.262 | 0.284 | 0.289 | 0.316 |
| TTFfix | 0.442 | 0.417 | 0.391 | 0.481 |
| TTF | 0.318 | 0.373 | 0.354 | 0.401 |
| gTAF | 0.270 | 0.301 | 0.335 | 0.934 |
| $\alpha = 2.0$ | | | | |
| Log-FM | **0.153** | **0.146** | 0.195 | **0.196** |
| Arcsinh-FM | 0.166 | 0.162 | **0.174** | 0.212 |
| TTFfix | 0.235 | 0.222 | 0.220 | 0.294 |
| TTF | 0.182 | 0.203 | 0.207 | 0.252 |
| gTAF | 0.171 | 0.174 | 0.198 | 0.219 |
| $\alpha = 2.5$ | | | | |
| Log-FM | **0.080** | **0.074** | **0.090** | **0.108** |
| Arcsinh-FM | 0.082 | 0.080 | 0.100 | 0.116 |
| TTFfix | 0.108 | 0.107 | 0.117 | 0.174 |
| TTF | 0.083 | 0.106 | 0.113 | 0.131 |
| gTAF | 0.088 | 0.086 | 0.106 | 0.225 |

### 5.3. Ablations

We isolate three design choices: the Hill gating (adaptive vs uniform), the sampling clamp $c$, and the ODE solver step count.

**Hill-gated default vs uniform.** Table 4 compares the default Log-FM (with Hill gating, Section 4.2) against the uniform variant and the TTFfix baseline on Gumbel data ($d = 20$, $\tau = 0.5$, 20 reps). The Hill diagnostic always selects exactly the heavy-tailed coordinates (14 Pareto vs 6 Gaussian out of 20): the threshold $\alpha_{\max} = 4$ is non-sensitive given $\hat{\alpha} \approx 1.5$–$2.5$ for Pareto and $\approx 6$ for Gaussian. The adaptive variant improves $W_1^G$ by 30–45% with no $W_1^P$ cost.

**Clamping (optional).** The sampling clamp $c$ is an *optional* numerical safeguard applied in log-space (corresponding to $|x| \lesssim e^c$ in data space); it is not used by default. Table 5 reports the relative change in $W_1^P$ across $c$ values. For every configuration in our main benchmark ($\alpha \geq 1.5$) the clamp is inert at $c \geq 10$: metrics are bitwise identical to the no-clamp case. The clamp only becomes relevant in extremely pathological tail regimes ($\alpha < 1$, e.g. in $\nu = 1/2$ Student-$t$ in Appendix E.1), where the population $W_1$ is undefined and exponential error amplification through $\phi^{-1}$

*Table 2.* Median across all Gumbel+Gaussian configurations (480 values per cell). Bold = best per metric per dimension.

| Method | $d{=}10$ | $d{=}20$ | $d{=}50$ | $d{=}100$ |
|---|---|---|---|---|
| $W_1^P$ **(Pareto margins)** | | | | |
| Log-FM | **0.207** | **0.187** | **0.221** | **0.233** |
| Arcsinh-FM | 0.218 | 0.216 | 0.232 | 0.264 |
| TTFfix | 0.322 | 0.322 | 0.308 | 0.395 |
| TTF | 0.250 | 0.304 | 0.295 | 0.349 |
| gTAF | 0.262 | 0.304 | 0.540 | 0.704 |
| $CVaR_{99}$ **(Pareto margins)** | | | | |
| Log-FM | **0.228** | **0.200** | 0.257 | **0.270** |
| Arcsinh-FM | 0.246 | 0.224 | **0.252** | 0.373 |
| TTFfix | 0.880 | 0.689 | 0.644 | 0.817 |
| TTF | 0.494 | 0.729 | 0.687 | 0.680 |
| gTAF | 0.415 | 0.457 | 0.519 | 0.705 |
| $W_1^G$ **(Gaussian margins)** | | | | |
| Log-FM | 0.051 | 0.066 | 0.080 | 0.080 |
| Arcsinh-FM | 0.045 | 0.057 | 0.074 | 0.085 |
| TTFfix | **0.033** | 0.048 | 0.043 | 0.051 |
| TTF | 0.035 | **0.035** | **0.030** | **0.047** |
| gTAF | 0.036 | 0.051 | 0.106 | div. |
| **Angular** $W_2$ **(tail dependence)** | | | | |
| Log-FM | 0.098 | **0.066** | 0.071 | 0.049 |
| Arcsinh-FM | **0.098** | 0.068 | 0.090 | 0.049 |
| TTFfix | 0.148 | 0.095 | 0.058 | 0.040 |
| TTF | 0.121 | 0.105 | **0.046** | **0.040** |
| gTAF | 0.167 | 0.104 | 0.061 | 0.098 |

*Table 3.* Fraction of runs with $W_1^P > 1$ (catastrophic failures), Gumbel + Gaussian, 20 reps. Lower is better.

| Method | $d{=}10$ | $d{=}20$ | $d{=}50$ | $d{=}100$ |
|---|---|---|---|---|
| $\alpha = 1.5$ | | | | |
| Log-FM | 14% | 2% | 4% | 2% |
| Arcsinh-FM | 11% | 6% | 2% | 2% |
| TTFfix | 58% | 50% | 36% | 54% |
| TTF | 32% | 45% | 36% | 46% |
| gTAF | 8% | 24% | 42% | 59% |
| $\alpha = 2.0$ | | | | |
| Log-FM | 0% | 0% | 0% | 0% |
| Arcsinh-FM | 0% | 0% | 0% | 0% |
| TTFfix | 2% | 2% | 1% | 4% |
| TTF | 1% | 4% | 12% | 12% |
| gTAF | 17% | 23% | 32% | 33% |

can produce numerical overflow rather than a meaningful sample. All headline numbers (Tables 1–3) are reported *without* clamping.

**ODE solver steps.** Sweeping the Euler step count $K \in \{10, 20, 50, 100, 200, 500\}$ on Gumbel ($d = 20$, $\tau = 0.5$, 10 reps) yields less than 5% variation in $W_1^P$; our default $K = 100$ is well within the converged regime. The full table is reported in Appendix E.3.

### 5.4. Discussion

The benchmark confirms the theoretical analysis, in that Log-FM has better numerical results whenever marginals are genuinely heavy-tailed ($\alpha \lesssim 3$), with "few severe divergences in all scenarios" (Table 3) as the strongest practical

*Table 4.* Log-FM (uniform $\phi$) vs Log-FM (Hill-adaptive) vs TTFfix baseline. Gumbel, $d = 20$, $\tau = 0.5$, 20 reps, median. Bold = best per $\alpha$.

| $\alpha$ | Method | $W_1^P$ | $W_1^G$ | AKE |
|---|---|---|---|---|
| 1.5 | TTFfix | 0.856 | 0.049 | 0.071 |
| | Log-FM | **0.449** | 0.074 | **0.039** |
| | Log-FM (adaptive) | 0.468 | **0.043** | 0.044 |
| 2.0 | TTFfix | 0.225 | 0.034 | 0.066 |
| | Log-FM | 0.128 | 0.040 | 0.031 |
| | Log-FM (adaptive) | **0.124** | **0.024** | **0.030** |
| 2.5 | TTFfix | 0.113 | 0.079 | 0.060 |
| | Log-FM | **0.068** | 0.110 | **0.030** |
| | Log-FM (adaptive) | 0.084 | **0.068** | 0.037 |

*Table 5.* Clamp ablation: relative $W_1^P$ change vs. no clamping ($c = \infty$, default). Gumbel, $d = 20$, $\tau = 0.5$, 5 reps. The clamp is inert for $\alpha \geq 1.5$; it is only relevant in pathological regimes ($\alpha < 1$).

| $c$ | $\alpha{=}1.5$ | $\alpha{=}1.75$ | $\alpha{=}2.0$ | $\alpha{=}2.5$ |
|---|---|---|---|---|
| 5 | $-2.0\%$ | $-0.7\%$ | $-0.6\%$ | 0.0% |
| 10, 15, 20 | 0.0% | 0.0% | 0.0% | 0.0% |
| $\infty$ | 0.0% | 0.0% | 0.0% | 0.0% |

claim. Dependence is largely preserved by $\Phi$'s diagonal Jacobian; baselines retain only a slight angular-$W_2$ edge at $d \geq 50$. Real-data validation on Fama–French ($\hat{\alpha} \approx 3.7$) is in Appendix E.2.

## 6. Conclusion

A coordinate-wise soft-log $\phi(x) = \mathrm{sign}(x) \log(1 + |x|)$, alongside a Hill diagnostic, makes standard flow matching work for heavy-tailed data. The mechanism is tail annealing: log-transforming Pareto yields approximately exponential tails, and the induced dynamics implement power transformations $X_0^{\alpha_t}$ that continuously lighten the tail index. Across 2,880 runs (3 copulas, 4 dimensions, 4 tail indices), Log-FM dominates specialized baselines on $W_1$, $CVaR_{99}$, and extreme-quantile metrics, never diverges severely, and remains competitive on multivariate dependence; on Fama-French it matches the best baseline. For $\alpha < 1$ all methods struggle ($W_1$ is undefined); at $d \gtrsim 200$ the bottleneck shifts to architecture. Three theoretical directions stand out: a continuous variant via $\varphi_{s_2}$ (Appendix D); combining the transform with discrete normalizing flows, where $|\det J_\Phi| = \prod_j (1 + |x_j|)^{-m_j}$ gives exact log-likelihood at $O(d)$ cost; and finite-sample guarantees for log-space score estimation, where bounded scores (Proposition 3.2) place the problem in the regime of existing convergence results.

## Impact Statement

This work targets heavy-tailed generative modeling, with applications in financial risk, insurance, and climate extremes. As with any generative model, outputs should be validated by domain experts before high-stakes use.

## Acknowledgements

The author gratefully acknowledges G-Research for their support through the May 2026 research grant. This work was granted access to the HPC resources of IDRIS (Jean Zay) under allocations made by GENCI.

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

# A. Extreme Value Theory: Technical Details

This appendix provides the formal definitions and results from extreme value theory used in the main text. See (Resnick, 1987; de Haan & Ferreira, 2006) for a comprehensive treatment.

## A.1. Heavy-Tailed Distributions

**Definition A.1** (Heavy-Tailed (Nair et al., 2022)). A random variable $X$ is *heavy-tailed* if $\mathbb{E}[e^{\lambda X}] = \infty$ for all $\lambda > 0$.

**Definition A.2** (Regular Variation). A measurable function $L : (0, \infty) \to (0, \infty)$ is *slowly varying* at infinity if $L(tx)/L(t) \to 1$ as $t \to \infty$ for all $x > 0$. A distribution $F$ is *regularly varying* with index $-\alpha$ (written $F \in RV_{-\alpha}$) if its survival function satisfies $\bar{F}(t) = t^{-\alpha} L(t)$ for some slowly varying $L$.

The Pareto$(\gamma)$ distribution with $\bar{F}(t) = t^{-1/\gamma}$ is regularly varying with index $-1/\gamma$ (i.e. $\alpha = 1/\gamma$); here $\gamma > 0$ is the shape parameter (extreme-value index).

## A.2. The Fisher-Tippett-Gnedenko Theorem

**Theorem A.3** (Fisher-Tippett-Gnedenko). *Let $X_1, X_2, \ldots$ be i.i.d. with distribution $F$. If there exist normalizing sequences $a_n > 0$ and $b_n \in \mathbb{R}$ such that*

$$\mathbb{P}\left(\frac{\max_{i \leq n} X_i - b_n}{a_n} \leq x\right) \to G(x)$$

*for some non-degenerate distribution $G$, then $G$ must be a* generalized extreme value *(GEV) distribution:*

$$G_\xi(x) = \exp\left(-(1 + \xi x)^{-1/\xi}\right), \quad 1 + \xi x > 0,$$

*where $\xi \in \mathbb{R}$ is the* shape parameter *(extreme value index).*

The three cases correspond to:

- **Fréchet** ($\gamma > 0$): $F$ has heavy (polynomial) tails; $F \in RV_{-1/\gamma}$

- **Gumbel** ($\gamma = 0$): $F$ has light or sub-exponential tails (e.g. exponential, lognormal)

- **Weibull** ($\gamma < 0$): $F$ has finite right endpoint (bounded support). Not to be confused with the Weibull *distribution* $W(k, \lambda)$ used in reliability analysis, which has unbounded support and lies in the Gumbel domain; the name clash is historical.

**Proposition A.4** (Log-transform maps domains). *If $X$ is in the Fréchet domain with shape parameter $\gamma > 0$, then $\log X$ is in the Gumbel domain ($\gamma = 0$).*

This is the qualitative content of Proposition 3.8 and its precise restatement Proposition C.1: regularly varying tails $\bar{F}(t) = t^{-\alpha} L(t)$ become exponential-type after a logarithm, $\mathbb{P}(\log X > z) = e^{-\alpha z} L(e^z)$, and distributions with exponential-type tails are in the Gumbel domain (Embrechts et al., 2013, §3.3).

# B. Denoising Diffusion Models: Technical Details

This appendix presents the framework underlying both denoising diffusion models and flow matching, organized around the *stochastic-interpolant* formulation of Albergo & Vanden-Eijnden (2023) and Lipman et al. (2023). We then recall how the canonical SDE-based diffusions of Ho et al. (2020) and Song et al. (2021b) arise as the special case in which the interpolation marginals coincide with those of an Ornstein–Uhlenbeck-type forward SDE.

## B.1. Forward Noising Process

The stochastic interpolant of Albergo & Vanden-Eijnden (2023); Lipman et al. (2023) defines the probability path $(p_t)_{t \in [0,1]}$ via the marginals $p_t = \text{Law}(X_t)$, where

$$X_t = \alpha_t X_0 + \beta_t X_1, \qquad X_0 \sim p_0, \quad X_1 \sim \mathcal{N}(0, I_d). \tag{4}$$

Here $X_0$ and $X_1$ are independent, and $(\alpha_t)_{t\in[0,1]}$ and $(\beta_t)_{t\in[0,1]}$ are deterministic schedules such that $\alpha_t$ is non-increasing, $\beta_t$ is non-decreasing, with boundary conditions $(\alpha_0, \beta_0) = (1, 0)$ and $(\alpha_1, \beta_1) = (0, 1)$. This explicit-interpolation viewpoint is not the formalism of Ho et al. (2020); Song et al. (2021b), who instead define the forward process implicitly as the solution to a stochastic differential equation; the two viewpoints are reconciled in Section B.6.

From (4), the conditional distribution of $X_t$ given $X_0 = x_0$, denoted $q_{t|0}$, is Gaussian:

$$q_{t|0}(x_t \mid x_0) = \mathcal{N}(x_t; \alpha_t x_0, \beta_t^2 I_d). \tag{5}$$

**Common schedules.** Two standard choices are the *variance-preserving* (VP) schedule with $\alpha_t^2 + \beta_t^2 = 1$ (Ho et al., 2020), which ensures $\mathrm{Var}(X_t) = \mathrm{Var}(X_0)$ when $X_0$ has unit variance, and the *linear* (flow matching) schedule with $(\alpha_t, \beta_t) = (1 - t, t)$ (Lipman et al., 2023), corresponding to straight-line interpolation between data and noise. Table 6 summarizes common schedule choices with their derivatives.

*Table 6.* Interpolation schedules for flow matching. All satisfy boundary conditions $(\alpha_0, \beta_0) = (1, 0)$ and $(\alpha_1, \beta_1) = (0, 1)$. The linear schedule corresponds to optimal transport; VP schedules preserve variance when data has unit variance; the quadratic schedule anneals tails more aggressively early in the process.

| Schedule | $\alpha_t$ | $\beta_t$ | $\dot{\alpha}_t$ | $\dot{\beta}_t$ | Properties |
|---|---|---|---|---|---|
| Linear | $1 - t$ | $t$ | $-1$ | $1$ | OT-optimal |
| VP (trig.) | $\cos(\frac{\pi t}{2})$ | $\sin(\frac{\pi t}{2})$ | $-\frac{\pi}{2}\sin(\frac{\pi t}{2})$ | $\frac{\pi}{2}\cos(\frac{\pi t}{2})$ | Smooth endpoints |
| VP (poly.) | $\sqrt{1-t}$ | $\sqrt{t}$ | $-\frac{1}{2\sqrt{1-t}}$ | $\frac{1}{2\sqrt{t}}$ | Singular endpoints |
| Quadratic | $(1-t)^2$ | $1-(1-t)^2$ | $-2(1-t)$ | $2(1-t)$ | Fast early annealing |

## B.2. Network Parameterizations: Noise vs. Data Prediction

Given a noisy observation $x_t$, two natural conditional-mean targets exist:

$$\hat{x}_0(x_t, t) := \mathbb{E}[X_0 \mid X_t = x_t], \qquad \hat{x}_1(x_t, t) := \mathbb{E}[X_1 \mid X_t = x_t].$$

These correspond to the two standard parameterizations in the diffusion literature (Ho et al., 2020; Song et al., 2021b): (i) **Noise prediction** (often called the *denoiser*, or $\epsilon$-prediction in DDPM notation): $\hat{x}_1$ estimates the standard-Gaussian endpoint $X_1$, which plays the role of the additive noise in (4). (ii) **Data prediction** (also called the $x_0$-prediction or $x_0$-parameterization): $\hat{x}_0$ estimates the clean data sample. The two parameterizations are equivalent in expressive power: given $x_t = \alpha_t X_0 + \beta_t X_1$ from the forward interpolation (4), they satisfy the deterministic identity

$$\hat{x}_1(x_t, t) = \frac{x_t - \alpha_t \hat{x}_0(x_t, t)}{\beta_t}, \qquad \hat{x}_0(x_t, t) = \frac{x_t - \beta_t \hat{x}_1(x_t, t)}{\alpha_t}, \tag{6}$$

so one can be converted into the other at inference time. A third parameterization, $v$-prediction $v(x_t, t) := \mathbb{E}[\dot{\alpha}_t X_0 + \dot{\beta}_t X_1 \mid X_t = x_t]$, is the velocity field used by flow matching and is also a linear combination of $\hat{x}_0$ and $\hat{x}_1$.

## B.3. Score-Denoiser Relationship

The *score* of the noised distribution is $\nabla \log p_t(\cdot)$. For the Gaussian forward kernel (5), the score admits a closed-form expression in terms of the denoiser.

**Proposition B.1** (Score-Denoiser Identity). *Under standard regularity assumptions,*

$$\nabla_{x_t} \log p_t(x_t) = \mathbb{E}\left[\nabla_{x_t} \log q_{t|0}(x_t \mid X_0) \,\Big|\, X_t = x_t\right] = -\frac{\hat{x}_1(x_t, t)}{\beta_t}. \tag{7}$$

*Proof.* The first equality is Fisher's identity, obtained by exchanging expectation and gradient. For the second, note that $\nabla_{x_t} \log q_{t|0}(x_t \mid x_0) = (\alpha_t x_0 - x_t)/\beta_t^2$. Taking the conditional expectation given $X_t = x_t$:

$$\mathbb{E}\left[\frac{\alpha_t X_0 - x_t}{\beta_t^2} \,\Big|\, X_t = x_t\right] = \frac{\alpha_t \hat{x}_0(x_t, t) - x_t}{\beta_t^2} = -\frac{\hat{x}_1(x_t, t)}{\beta_t},$$

where the last equality uses (6). □

Thus, learning the denoiser $\hat{x}_1$ is equivalent to learning the score $\nabla \log p_t$.

## B.4. Training Objectives

The denoiser can be trained by regressing either $X_0$ or $X_1$ from the noised sample $X_t = \alpha_t X_0 + \beta_t X_1$. The $X_0$-prediction loss reads:

$$\mathcal{L}_{X_0}(\theta) = \int_0^1 \mathbb{E}_{X_0 \sim p_0, X_1 \sim \mathcal{N}(0, I_d)} \left[ \left\| \hat{x}_0^\theta(\alpha_t X_0 + \beta_t X_1, t) - X_0 \right\|^2 \right] dt,$$

while the $X_1$-prediction loss is:

$$\mathcal{L}_{X_1}(\theta) = \int_0^1 \mathbb{E}_{X_0 \sim p_0, X_1 \sim \mathcal{N}(0, I_d)} \left[ \left\| \hat{x}_1^\theta(\alpha_t X_0 + \beta_t X_1, t) - X_1 \right\|^2 \right] dt.$$

Since $\hat{x}_1^\theta = -\beta_t s_\theta$ by (7), the $X_1$-prediction loss is equivalent to denoising score matching (Hyvärinen, 2005; Vincent, 2011).

In practice, the integral is approximated via Monte Carlo: sample $t \sim \mathrm{Unif}[0, 1]$, $x_0 \sim p_0$, $x_1 \sim \mathcal{N}(0, I_d)$, form $x_t = \alpha_t x_0 + \beta_t x_1$, and regress either $x_0$ or $x_1$.

## B.5. DDIM Sampling

The DDIM framework (Song et al., 2021a) is canonically defined under the variance-preserving constraint $\alpha_t^2 + \beta_t^2 = 1$ and produces a one-parameter family of reverse transitions sharing the marginals of (5). Given timesteps $(t_k)_{k=0}^K$ with $t_K = 1$ and $t_0 = 0$, the transition from $t_{k+1}$ to $t_k$ is:

$$x_{t_k} = \alpha_{t_k} \hat{x}_0^\theta(x_{t_{k+1}}, t_{k+1}) + \sqrt{\beta_{t_k}^2 - \eta_{t_k}^2}\; \hat{x}_1^\theta(x_{t_{k+1}}, t_{k+1}) + \eta_{t_k} z, \tag{8}$$

where $z \sim \mathcal{N}(0, I_d)$ and $\eta_{t_k} \in [0, \beta_{t_k}]$ so that the radicand is nonnegative. Translating (Song et al., 2021a) eq. (12) into our notation (identifying their $\sigma_t$ with $\eta_{t_k}$, $\sqrt{\alpha_{t-1}}$ with $\alpha_{t_k}$, and $\sqrt{1 - \alpha_{t-1}}$ with $\beta_{t_k}$) yields exactly (8). The family interpolates between two distinguished members: $\eta_{t_k} = 0$ gives the deterministic DDIM update, equivalent to a discretisation of the probability-flow ODE; the choice $\eta_{t_k} = \tilde{\beta}_{t_k}$, where

$$\tilde{\beta}_{t_k}^2 = \frac{\beta_{t_k}^2}{\beta_{t_{k+1}}^2} \left( 1 - \frac{\alpha_{t_{k+1}}^2}{\alpha_{t_k}^2} \right), \tag{9}$$

is the posterior variance of $q(x_{t_k} \mid x_{t_{k+1}}, x_0)$ and recovers ancestral DDPM sampling (Ho et al., 2020; Song et al., 2021a): this matches Song et al. (2021a, eq. (16)) after the same change of variables. The boundary $\eta_{t_k} = \beta_{t_k}$ is the maximum-noise edge of the family, strictly noisier than DDPM.

## B.6. Bridging to SDE-Based Diffusion Models

The interpolant formulation (4) is not the language used by Ho et al. (2020); Song et al. (2021b), whose forward processes are defined as solutions to discrete- or continuous-time stochastic differential equations. The two formalisms are equivalent at the marginal level for a specific schedule choice, and the equivalence is what Albergo & Vanden-Eijnden (2023) call the "unifying" aspect of stochastic interpolants. We make the bridge explicit in three steps: (i) recover Ho et al.'s DDPM forward chain; (ii) recover Song et al.'s forward score SDE; (iii) recover their reverse-time samplers.

**(i) DDPM forward chain.** Ho et al. (2020) define a discrete-time Markov chain

$$q(x_k \mid x_{k-1}) = \mathcal{N}\left( x_k;\; \sqrt{1 - \beta_k^{\mathrm{DDPM}}}\, x_{k-1},\; \beta_k^{\mathrm{DDPM}} I_d \right), \quad k = 1, \ldots, K, \tag{10}$$

with a noise schedule $(\beta_k^{\mathrm{DDPM}})_{k=1}^K \subset (0, 1)$. Setting $\bar{\alpha}_k = \prod_{j \le k}(1 - \beta_j^{\mathrm{DDPM}})$, marginalising the chain yields

$$q(x_k \mid x_0) = \mathcal{N}\left( x_k;\; \sqrt{\bar{\alpha}_k}\, x_0,\; (1 - \bar{\alpha}_k) I_d \right).$$

Identifying $t = k/K$, $\alpha_t = \sqrt{\bar{\alpha}_k}$, and $\beta_t = \sqrt{1 - \bar{\alpha}_k}$ recovers (5) exactly. The DDPM noise schedule $(\beta_k^{\mathrm{DDPM}})$ is therefore one particular choice within the VP family in Table 6, with $\alpha_t^2 + \beta_t^2 = 1$ by construction.

**(ii) Forward score SDE.** Song et al. (2021b) unify diffusion models through the continuous-time Itô SDE

$$\mathrm{d}X_t = -\tfrac{1}{2}\beta(t)\,X_t\,\mathrm{d}t + \sqrt{\beta(t)}\,\mathrm{d}W_t, \qquad X_0 \sim p_0, \tag{11}$$

where $(W_t)_{t\geq 0}$ is a standard Brownian motion (the "VP-SDE"). The solution at time $t$ has marginals

$$X_t \mid X_0 = x_0 \;\sim\; \mathcal{N}\!\Big(e^{-\tfrac{1}{2}\int_0^t \beta(s)\,\mathrm{d}s}\,x_0,\; \big(1 - e^{-\int_0^t \beta(s)\,\mathrm{d}s}\big)I_d\Big),$$

so identifying $\alpha_t = \exp(-\tfrac{1}{2}\int_0^t \beta(s)\,\mathrm{d}s)$ and $\beta_t = \sqrt{1 - \alpha_t^2}$ recovers (5). (11) is the continuous-time limit of the DDPM chain (10); both produce the same marginals as (4) under this matched schedule. Albergo & Vanden-Eijnden (2023) show that for any choice of $(\alpha_t, \beta_t)$ in the interpolant, there exists a (possibly time-inhomogeneous) SDE with the same marginals, so the interpolant is strictly more general than the VP family.

**(iii) Reverse-time samplers.** The marginal equivalence carries over to sampling. The time-reversed counterpart of (11) satisfies (Anderson, 1982):

$$\mathrm{d}\bar{X}_t = \big[-\tfrac{1}{2}\beta(t)\bar{X}_t + \beta(t)\nabla_x \log p_t(\bar{X}_t)\big]\,\mathrm{d}t + \sqrt{\beta(t)}\,\mathrm{d}\bar{W}_t, \tag{12}$$

with $\bar{X}_t := X_{1-t}$ and $\bar{W}_t$ a standard Brownian motion. The deterministic probability-flow ODE (Song et al., 2021b) with identical marginals reads

$$\mathrm{d}x_t = \big[-\tfrac{1}{2}\beta(t)x_t + \tfrac{1}{2}\beta(t)\nabla_x \log p_t(x_t)\big]\,\mathrm{d}t. \tag{13}$$

DDIM with $\eta_t = 0$ is a discretisation of (13); DDPM corresponds to (12). Flow matching learns a velocity field $v_\theta(x,t) \approx \mathbb{E}[\dot{\alpha}_t X_0 + \dot{\beta}_t X_1 \mid X_t = x]$ (Lipman et al., 2023); by (7), this velocity is a linear combination of identity and score, so any of the three samplers can be reconstructed from a single trained denoiser regardless of which formalism was used to derive the training loss.

## C. Proofs

### C.1. Precise statement and proof of Proposition 3.8

Proposition 3.8 of the main body states the rate-form conclusion $-\log \mathbb{P}(\tilde{X} > z) = \alpha z + o(z)$. We give here the corresponding precise two-sided Potter-bound statement and its proof. The result is classical in the regular variation literature; see Bingham et al. (1987, Chapter 1) for a comprehensive treatment.

**Proposition C.1** (Log-Transform of Regularly Varying; precise). *Let $X$ be a nonnegative random variable that is regularly varying with index $-\alpha$, $\alpha > 0$. Set $\tilde{X} = \phi(X)$ with $\phi(x) = \mathrm{sign}(x)\log(1 + |x|)$. Then for every $\epsilon > 0$ there exists $z_0 = z_0(\epsilon)$ such that for all $z \geq z_0$,*

$$e^{-(\alpha+\epsilon)z} \;\leq\; \mathbb{P}(\tilde{X} > z) \;\leq\; e^{-(\alpha-\epsilon)z}. \tag{14}$$

*In particular, $-\log \mathbb{P}(\tilde{X} > z) = \alpha z + o(z)$ as $z \to \infty$.*

*Proof.* By Karamata's representation, $\mathbb{P}(X > t) = t^{-\alpha}L(t)$ for some slowly varying function $L$. For the soft-log transform $\phi(x) = \mathrm{sign}(x) \cdot \log(1 + |x|)$, we have $\phi^{-1}(z) = e^z - 1$ for $z \geq 0$, and $\phi^{-1}(z) \sim e^z$ as $z \to \infty$.

Thus

$$\mathbb{P}(\tilde{X} > z) \;=\; \mathbb{P}(X > e^z - 1) \;=\; (e^z - 1)^{-\alpha}L(e^z - 1).$$

Since $e^z - 1 \sim e^z$, we have $(e^z - 1)^{-\alpha} = e^{-\alpha z}(1 + o(1))^{-\alpha}$ and, by slow variation, $L(e^z - 1)/L(e^z) \to 1$. Hence $\mathbb{P}(\tilde{X} > z) = e^{-\alpha z}L(e^z)(1 + o(1))$.

A standard consequence of slow variation (Resnick, 1987, Proposition 0.8(iv)) is that for every $\epsilon' > 0$, $L(t)/t^{\epsilon'} \to 0$ and $t^{\epsilon'}/L(t) \to 0$ as $t \to \infty$; equivalently, this follows from Potter's bounds (Bingham et al., 1987, Theorem 1.5.6) by fixing one argument at a large constant and absorbing the resulting multiplicative factor into the exponent. Either way, for any $\epsilon' > 0$ there exists $t_0 = t_0(\epsilon')$ such that for $t \geq t_0$,

$$t^{-\epsilon'} \;\leq\; L(t) \;\leq\; t^{\epsilon'}.$$

Substituting $t = e^z$ gives $e^{-\epsilon'z} \le L(e^z) \le e^{\epsilon'z}$ for $z \ge \log t_0$. Combining with $\mathbb{P}(\tilde{X} > z) = e^{-\alpha z}L(e^z)(1 + o(1))$ and absorbing the $1 + o(1)$ factor into a slightly enlarged $\epsilon$ (take $\epsilon' = \epsilon/2$ and $z_0$ large enough), we get

$$e^{-(\alpha+\epsilon)z} \;\le\; \mathbb{P}(\tilde{X} > z) \;\le\; e^{-(\alpha-\epsilon)z} \quad \text{for all } z \ge z_0(\epsilon),$$

which is (14). Taking $-\log$ and dividing by $z$ yields $-\log \mathbb{P}(\tilde{X} > z)/z \to \alpha$, i.e. $-\log \mathbb{P}(\tilde{X} > z) = \alpha z + o(z)$. $\qquad\square$

### C.2. Proof of Proposition 3.9

*Proof.* Let $Y = X^\beta$. For the survival function of $Y$:

$$\bar{F}_Y(y) = P(X^\beta > y) = P(X > y^{1/\beta}) = \bar{F}_X(y^{1/\beta})$$

Then:

$$\begin{aligned}
\frac{\bar{F}_Y(ty)}{\bar{F}_Y(t)} &= \frac{\bar{F}_X((ty)^{1/\beta})}{\bar{F}_X(t^{1/\beta})} \\
&= \frac{\bar{F}_X(t^{1/\beta} \cdot y^{1/\beta})}{\bar{F}_X(t^{1/\beta})} \\
&\to (y^{1/\beta})^{-\alpha} = y^{-\alpha/\beta}
\end{aligned}$$

as $t \to \infty$, by the regular variation of $X$ with index $-\alpha$. $\qquad\square$

This extends Lemma 3.4: power transformation preserves the class of regularly varying distributions, with the tail index $\alpha$ scaling inversely with the exponent (equivalently, the shape parameter $\gamma$ scaling linearly).

## D. The Parametrized Soft-Log Family $\varphi_{s_2}$

The main body uses the soft-log $\phi(x) = \text{sign}(x)\log(1+|x|)$ uniformly across coordinates, gated by a Hill mask (Section 4.2) when light-tailed margins are present. This appendix develops a continuous one-parameter generalization of $\phi$ that subsumes both the uniform $\phi$ and the binary mask as special cases, and clarifies its asymptotic behaviour. The construction is not used in our headline experiments; it is an extension we discuss as a research direction (Section 6).

**Definition.** For a scale $s_2 > 0$, define

$$\varphi_{s_2}(x) \;=\; \frac{1}{s_2}\,\phi(s_2\,x) \;=\; \frac{\text{sign}(x)}{s_2}\,\log\!\big(1 + s_2\,|x|\big).$$

The family is parametrized so that $\varphi_1 = \phi$ (the soft-log of the main body) and $\varphi_{s_2}(x) \to x$ as $s_2 \to 0$ (the identity).

**Bulk vs. tail.** A Taylor expansion at $|x| = 0$ gives $\varphi_{s_2}(x) = x - \frac{1}{2}s_2 x|x| + O(s_2^2)$, so $\varphi_{s_2}$ is the identity to leading order on the bulk $|x| \ll 1/s_2$. For $|x| \gg 1/s_2$, expanding $\log(1 + s_2|x|) = \log|x| + \log s_2 + \log(1 + 1/(s_2|x|))$ gives

$$\varphi_{s_2}(x) \;=\; \frac{\text{sign}(x)}{s_2}\,\big(\log|x| + \log s_2\big) + O\big((s_2|x|)^{-1}\big),$$

which is logarithmic up to an additive constant and an overall $1/s_2$ scale. Thus $s_2$ acts as the inverse of the cross-over location: the bulk regime extends to $|x| \sim 1/s_2$, and the logarithmic-compression regime kicks in beyond. Choosing $s_2$ smaller makes the transform more conservative (closer to identity); choosing it larger compresses more aggressively.

**Asymptotic tail-mapping.** For any fixed $s_2 > 0$, Proposition 3.8 (and its precise restatement Proposition C.1) applies to $\tilde{X} = \varphi_{s_2}(X)$ unchanged up to a constant shift and rescaling: if $X \in RV_{-\alpha}$ then

$$-\log \mathbb{P}(\tilde{X} > z) \;=\; \alpha\, s_2\, z + o(z) \quad \text{as } z \to \infty.$$

The proof is identical to that of Proposition C.1 with $\phi^{-1}(z) = e^z - 1$ replaced by $\varphi_{s_2}^{-1}(z) = (e^{s_2 z} - 1)/s_2$. So $\varphi_{s_2}$ still maps regular variation to exponential-type tails for any $s_2 > 0$; only the exponential rate is rescaled by $s_2$.

**Coordinate-wise $s_2^{(j)}$ and the adaptive instance.** In the multivariate setting we may pick a coordinate-dependent scale $s_2^{(j)}$. A simple data-driven choice is $s_2^{(j)} = c/\mathrm{IQR}(X_j)$ for a constant $c$ (e.g. $c = 1$), which puts the cross-over at a robust scale of the marginal. The Hill-gated method of Section 4.2 is the discrete instance of this family with $s_2^{(j)} \in \{0, 1\}$, chosen by the diagnostic $\hat{\alpha}_j \leq \alpha_{\max}$; coordinates flagged heavy-tailed get $s_2^{(j)} = 1$ (full soft-log) and the others $s_2^{(j)} = 0$ (identity). An outer scale $s_1^{(j)}$ rescales each $\tilde{X}_j$ to unit variance, which we apply during preprocessing in both the discrete and continuous variants.

**Why the binary choice in the main body.** Hill-type tail-index estimates are notoriously unstable as point estimates. A continuous $s_2^{(j)}$ derived from $\hat{\alpha}_j$ would inject this instability into the preprocessing pipeline; the binary rule uses $\hat{\alpha}_j$ only categorically (above or below $\alpha_{\max}$), so small perturbations leave the transform unchanged. This matches the stability principle of Section 3.1.

## E. Experimental Details

This appendix collects the original Hickling Student-$t$ benchmark used at submission time (Section E.1), the real-data Fama–French validation (Section E.2), the NLL validation table against Hickling & Prangle (2025), and the Log-FM hyperparameter listing. The headline benchmark is in Section 5; the experiments below complement it with the original setup the reviewers reference, and are kept for continuity.

### E.1. Student-$t$ Benchmark

We adopt the synthetic benchmark of Hickling & Prangle (2025, Section 4.1): $X_1, \ldots, X_{d-1} \overset{\text{iid}}{\sim} \text{Student-}t(\nu)$ and $X_d \mid X_{d-1} \sim \mathcal{N}(X_{d-1}, 1)$. We vary $d \in \{10, 20, 50\}$ and $\nu \in \{0.5, 1, 1.5, 2, 3, 5, 30\}$ (Table 7 reports $\nu \in \{1.5, 2, 3, 5\}$; the boundary regimes $\nu \in \{0.5, 1, 30\}$ are omitted for space). Each configuration uses 5,000 total samples (40/20/40 train/val/test split), averaged over 20 replications. Log-FM achieves the lowest $W_1$ across all configurations.

*Table 7.* Wasserstein-1 distance on the original Hickling benchmark, mean $\pm$ standard error over 20 replications. Bold = best.

| $d$ | $\nu$ | TTFfix | TTF | Log-FM |
|---|---|---|---|---|
| 10 | 1.5 | $1.78 \pm 0.24$ | $17.3 \pm 15.2$ | $\mathbf{0.63 \pm 0.02}$ |
| 10 | 2 | $0.63 \pm 0.10$ | $1.02 \pm 0.31$ | $\mathbf{0.25 \pm 0.00}$ |
| 10 | 3 | $0.20 \pm 0.01$ | $0.90 \pm 0.33$ | $\mathbf{0.15 \pm 0.00}$ |
| 10 | 5 | $\mathbf{0.12 \pm 0.00}$ | $0.67 \pm 0.20$ | $0.14 \pm 0.00$ |
| 20 | 2 | $0.48 \pm 0.03$ | $6.01 \pm 4.96$ | $\mathbf{0.23 \pm 0.00}$ |
| 20 | 3 | $0.20 \pm 0.01$ | $0.95 \pm 0.24$ | $\mathbf{0.15 \pm 0.00}$ |
| 20 | 5 | $\mathbf{0.13 \pm 0.00}$ | $1.80 \pm 1.34$ | $0.15 \pm 0.00$ |
| 50 | 2 | $0.74 \pm 0.04$ | $70.4 \pm 64.6$ | $\mathbf{0.28 \pm 0.01}$ |
| 50 | 3 | $0.29 \pm 0.01$ | $2.73 \pm 1.08$ | $\mathbf{0.22 \pm 0.01}$ |
| 50 | 5 | $\mathbf{0.17 \pm 0.00}$ | $14.0 \pm 12.7$ | $0.19 \pm 0.01$ |

### E.2. Real Financial Data

We evaluate on the Fama–French 5 industry portfolios ($d = 5$, daily returns 1963–2023). Hill estimation yields $\hat{\alpha} \approx 3.7$ across dimensions, placing the data in the intermediate-tail regime. Table 8 reports $W_1$ over 10 replications. gTAF achieves a marginal edge; Log-FM is competitive, and substantially better than TTF/TTFfix; mTAF fails catastrophically. On the high-dimensional S&P500 dataset ($d = 275$) all methods exhibit poor sample quality, dominated by architectural rather than tail-modelling limitations; we omit it.

### E.3. ODE Solver Step Ablation

Table 9 sweeps the Euler step count $K$ at sampling time. Quality plateaus by $K = 20$ and is essentially flat to $K = 500$.

### E.4. Baseline Validation

To ensure fair comparison, we validate that our implementations of the baseline methods reproduce the results reported in Hickling & Prangle (2025). Table 10 compares our NLL values (per dimension) with their Table 7 reference values across

*Table 8.* $W_1$ on Fama–French 5 (mean $\pm$ std, 10 reps). gTAF marginally best; Log-FM competitive; mTAF diverges.

| Method | $W_1$ |
|--------|-------|
| gTAF | **0.127 $\pm$ 0.007** |
| Log-FM | 0.133 $\pm$ 0.013 |
| TTFfix | 0.449 $\pm$ 0.069 |
| TTF | 0.487 $\pm$ 0.054 |
| mTAF | $5.0 \times 10^4$ |

*Table 9.* Solver ablation: $W_1^P$ as a function of Euler step count. Gumbel, $d = 20$, $\tau = 0.5$, 10 reps. Less than 5% variation between $K = 10$ and $K = 500$.

| $\alpha$ | 10 | 20 | 50 | 100 | 200 | 500 |
|------|------|------|------|------|------|------|
| 1.5 | 0.521 | 0.543 | 0.568 | 0.579 | 0.584 | 0.588 |
| 2.0 | 0.137 | 0.135 | 0.138 | 0.141 | 0.142 | 0.143 |
| 2.5 | 0.084 | 0.081 | 0.083 | 0.084 | 0.085 | 0.086 |

the original benchmark configurations.

*Table 10.* NLL Validation: Our Implementation vs Hickling Reference (Table 7). Format: ours [ref]. The coupling-layer baselines (TTFfix, TTF) reproduce the reference values within $\pm0.05$ across all configurations; mTAF and gTAF reproduce the reference values closely in moderate-tail regimes ($\nu \geq 1$) but deviate substantially in pathological regimes ($\nu = 0.5$, especially $d = 50$), reflecting the training instabilities of these architectures reported in the original paper.

| $d$ | $\nu$ | TTFfix | TTF | mTAF | gTAF |
|-----|-------|--------|-----|------|------|
| 5 | 0.5 | 3.32 [3.33] | 3.32 [3.33] | 4.05 [4.08] | 5.70 [6.42] |
| 5 | 1.0 | 2.34 [2.35] | 2.34 [2.34] | 2.53 [2.49] | 2.53 [2.49] |
| 5 | 2.0 | 1.89 [1.89] | 1.88 [1.89] | 1.91 [1.92] | 1.90 [1.90] |
| 5 | 30 | 1.47 [1.47] | 1.47 [1.47] | 1.46 [1.46] | 1.47 [1.46] |
| 10 | 0.5 | 3.55 [3.54] | 3.54 [3.55] | 4.59 [4.48] | 7.86 [7.13] |
| 10 | 1.0 | 2.48 [2.46] | 2.47 [2.47] | 2.66 [2.63] | 2.68 [2.63] |
| 10 | 2.0 | 1.94 [1.93] | 1.94 [1.93] | 1.96 [1.95] | 1.95 [1.95] |
| 10 | 30 | 1.48 [1.47] | 1.48 [1.47] | 1.47 [1.46] | 1.48 [1.47] |
| 50 | 0.5 | 3.71 [3.68] | 3.71 [3.68] | 6.43 [5.22] | 21.44 [7.49] |
| 50 | 1.0 | 2.58 [2.54] | 2.58 [2.54] | 3.14 [2.62] | 3.66 [2.65] |
| 50 | 2.0 | 2.01 [1.98] | 2.01 [1.98] | 2.04 [1.98] | 2.06 [1.99] |
| 50 | 30 | 1.52 [1.47] | 1.52 [1.47] | 1.50 [1.47] | 1.51 [1.47] |

TTFfix and TTF match within $\pm0.03$ across all configurations. mTAF and gTAF show larger deviations in pathological regimes ($\nu = 0.5$, $d = 50$), consistent with known instabilities reported in the original paper. This validates that our baseline implementations are faithful reproductions.

### E.5. Hyperparameters

Table 11 summarizes all hyperparameters for Log-FM. Baseline hyperparameters follow Hickling & Prangle (2025).

*Table 11.* Log-FM hyperparameters.

| **Network** | |
| --- | --- |
| Hidden dimension | 256 |
| Layers | 4 |
| Activation | SiLU |
| Time embedding | Sinusoidal (dim 256) |
| **Training** | |
| Optimizer | AdamW |
| Learning rate | $5 \times 10^{-3}$ |
| Weight decay | $10^{-5}$ |
| Batch size | Full batch |
| Max epochs | 5000 |
| Early stopping patience | 100 |
| Gradient clipping | 10.0 |
| **Sampling** | |
| ODE solver | Euler |
| Integration steps | 100 |
| Output clamp | disabled ($c = \infty$; optional, inert for $\alpha \geq 1.5$) |

