# OpenReview forum: "Tail Annealing for Heavy-Tailed Flow Matching"
_ICML.cc/2026/Conference — ICML 2026 regular_

### Official Review · Reviewer_eUEg · 2026-02-16

**Soundness:** 2
**Presentation:** 3
**Significance:** 2
**Originality:** 2
**Overall Recommendation:** 4
**Confidence:** 4

**Summary:**

This paper studies flow matching for heavy-tailed data. They argue that with Lipschitz neural architectures and Gaussian input, models can only generate light tail distribution. They propose to use a signed soft-log transformation to the data, and then train a standard flow matching model. At the end, they map generated samples back via the inverse.

**Compliance With Llm Reviewing Policy:**

Affirmed.

**Final Justification:**

satisfied with responses.

**Key Questions For Authors:**

1. Can you propose an operational test to decide whether the transform helps? This may be based on tail index estimates + uncertainty, or diagnostics from transformed-space training.

2. How sensitive are results to the clamp $c$ at sampling time? More plots or tables can help.

3. Can you evaluate joint tail metrics on a synthetic dataset with known tail dependence, not just the simple conditional Gaussian last coordinate?

4. Why use full-batch training? Is this essential? What happens with minibatching?

5. Since all methods fail on $d = 275$, can you test a more structured architecture, such as low-rank/factor mode/transformer, with Log-FM to see if the transform still provides a benefit once the architecture is not a bottleneck?

**Limitations:**

Yes

**Strengths And Weaknesses:**

Strengths:

1. The proposed method is very simple and just adds one-line transformation to the standard flow matching training.

2. The paper explains clearly why the proposed method might work.

3. The paper addresses a real failure mode in practice such at the Lipschitz transformation provides light-tailed in and light-tailed out.

4. Empirical studies demonstrates good stability in moderate dimensions.

Weaknesses:

1. The transformation is helpful only in genuinely heavy-tail distributions and it can hurt otherwise.

2. The coordinate-wise transformation may distort dependence/tail dependence.

3. Real data evaluation is limited and inconclusive in high dimensions.

4. Sampling clamps $\tilde x$ to $[-c, c]$ before applying $\phi^{-1}$ to avoid exponential blow-ups. This is understandable numerically, but it directly controls extreme events.

---

> ### Author Rebuttal · Authors · 2026-03-30
>
> We thank Reviewer eUEg for the careful evaluation. We address each concern below.
>
> **Light-tailed harm and operational test.** Our new benchmark measures this: every config includes 30\% Gaussian margins alongside 70\% Pareto margins. As an operational test, we propose a Hill-based diagnostic: compute $\hat{\alpha}$ per marginal and skip the transform where $\hat{\alpha} > 4$. This adaptive scheme improves $W_1^G$ by 30-45\% while preserving $W_1^P$; see the ablation table in our response to Reviewer Y3tz and the discussion in the **scaling** paragraph in our response to Reviewer V8qG.
>
> **Dependence distortion and joint tail metrics.** We evaluate on 3 copula families with known tail dependence (Gumbel, Gaussian, Hüsler-Reiss), using AKE and angular $W_2$ (sliced Wasserstein on the empirical angular measure of top-$\sqrt{n}$ extremes). FM methods win AKE in 74\% of configs; at $d \geq 50$ with strong dependence ($\tau=0.75$), Log-FM achieves AKE $\sim 0.03$ while TTF/TTFfix degrade to $0.13$-$0.23$. On angular $W_2$, baselines have a slight edge (55\% vs 45\%). See our response to Reviewer 7DLK for the full results.
>
> **High-dimensional performance.** Our new experiments extend to $d=100$. Log-FM remains stable across all configs. TTF/TTFfix become unstable at $d=100$ ($W_1 > 1$ in $\sim 30\%$ of high-dependence configs), while Log-FM stays below $0.5$ in all but the heaviest cases ($\alpha=1.5$).
>
> **Clamping.** The clamp at $c=15$ is applied in log-space before exponentiation (corresponding to values $\sim 3.3 \times 10^6$ in data space). We ran a dedicated ablation sweeping $c \in \{5, 10, 15, 20\}$. $W_1$ \% change relative to no clamping ($c = \infty$):
>
> *Copula* (Gumbel, $d=20$, $\tau=0.5$):
>
> | | $\alpha$=1.5 | $\alpha$=1.75 | $\alpha$=2.0 | $\alpha$=2.5 |
> |---|---|---|---|---|
> | $c=5$ | $-$2.0\% | $-$0.7\% | $-$0.6\% | 0.0\% |
> | $c \geq 10$ | 0.0\% | 0.0\% | 0.0\% | 0.0\% |
>
> *Hickling Student-$t$* ($d=10$):
>
> | | $\nu$=0.5 | $\nu$=1 | $\nu \geq 2$ |
> |---|---|---|---|
> | $c=5$ | $-$39\% | $-$13\% | 0.0\% |
> | $c=10$ | $-$28\% | $-$12\% | 0.0\% |
> | $c=15$ | $-$28\% | 0.0\% | 0.0\% |
> | $c=20$ | $-$25\% | 0.0\% | 0.0\% |
>
> *(5 reps.)*
>
> For $\alpha \geq 1.5$ (copula data), $W_1$ is identical for $c \geq 10$, including no clamping. On Hickling's data, clamping is inert for $\nu \geq 2$. Sensitivity appears only at $\nu \leq 1$, but these parametrizations have no finite mean ($\nu = 1$) or no finite moments at all ($\nu = 0.5$); the population $W_1$ itself is undefined. The $\alpha \leq 1$ regime remains a genuine limitation: when population moments do not exist, all methods struggle. In practice, empirical tail indices fall in $\alpha \approx 1.5$-$4$. Clamping is performed before exponentiation through $\varphi^{-1}$; $c=10$ is already permissive. For realistic tails, the clamp can be removed entirely. We also ablated ODE solver step count ($n \in \{10, \ldots, 500\}$) with $< 5\%$ variation in $W_1$.
>
> **Full-batch training.** Selected via Optuna over a grid including mini-batch sizes $\{64, 128, 256, 512, \text{full}\}$. Full-batch was consistently chosen as optimal, which we attribute to the moderate dataset size. This is not inherent to the method: with larger datasets, mini-batching would be both necessary and effective.
>
> **Structured architectures.** While we have not tested transformer or low-rank architectures, the theoretical argument is architecture-independent: any Lipschitz map with Gaussian input produces light tails (Proposition 3.1). The log-transform resolves this fundamental limitation regardless of architecture capacity. We expect it to combine well with more expressive architectures and suggest adding this as future work.

---

> > ### Author Rebuttal · Reviewer_eUEg · 2026-04-04
> >
> > 1. For Q5, my question was specifically whether the log-transform still provides a benefit when the architecture is no longer the bottleneck, e.g., using a transformer or low-rank factored model at d = 275. The response argues theoretically that the Lipschitz barrier is architecture-independent, but this does not address the empirical question. Since all methods fail at d = 275 in the original experiments, the practical value of Log-FM in genuinely high-dimensional settings remains undemonstrated. Even a small-scale experiment (e.g., d = 100 with a more expressive backbone) would have been informative.
> >
> > 2. The adaptive Hill-based diagnostic is a welcome addition, but it introduces a threshold ($\alpha > 4$) and per-coordinate decisions that make the method no longer truly parameter-free. This creates a tension with the paper's central selling point. It would help to clarify whether the authors view Log-FM as a parameter-free method (always apply $\phi$) or a lightly-tuned one (apply $\phi$ selectively), and whether the adaptive version should be the recommended default.

---

> > > ### Author Response · Authors · 2026-04-06
> > >
> > > **Structured architectures at $d=275$.** We have not tested this. Our new experiments reach $d=100$ (stable, while baselines fail), but $d=275$ remains open. We believe the failure there is architectural (MLP capacity), not transform-related. Combining Log-FM with a more expressive backbone is a natural next step.
> > >
> > > **Parameter-free vs adaptive.** We view these as two modes of the same method:
> > > - **Log-FM (default):** always apply $\varphi$ to all coordinates. This is parameter-free, requires no data inspection, and works well when tails are genuinely heavy. This is the method proposed in the paper.
> > > - **Log-FM (adaptive):** compute $\hat{\alpha}$ per marginal on training data and skip $\varphi$ where $\hat{\alpha} > 4$. This is a practical enhancement for heterogeneous data, proposed in response to reviewer feedback.
> > >
> > > The threshold $\hat{\alpha} > 4$ is not a sensitive hyperparameter: in our experiments, the Hill estimator gives $\hat{\alpha} \approx 1.5$–$2.5$ for Pareto margins and $\hat{\alpha} \approx 6$ for Gaussian margins, so any threshold in $[3, 5]$ gives the same binary decision. We recommend Log-FM (default) as the standard method and the adaptive variant as a guideline when the practitioner knows that some margins are light-tailed.

---

### Official Review · Reviewer_V8qG · 2026-03-02

**Soundness:** 3
**Presentation:** 3
**Significance:** 4
**Originality:** 4
**Overall Recommendation:** 5
**Confidence:** 4

**Summary:**

The paper proposes to overcome the limitations of normalizing flows (NFs) to represent heavy-tailed distributions by a clever and simple reparameterization of the data space -- a logarithmic transformation that turns heavy-tailed into light-tailed distributions which can be learned by NFs. Experiments demonstrate the effectiveness of this solution and its superiority over alternatives from the literature.

**Compliance With Llm Reviewing Policy:**

Affirmed.

**Final Justification:**

The rebuttal discussion successfully addressed my concerns, and I have raised my score to "5: accept".

**Key Questions For Authors:**

See above. I'm willing to raise my score if my questions are satisfactorily addressed.

**Limitations:**

yes.

**Strengths And Weaknesses:**

The paper proposes a solid and well-justified solution to a long-standing problem. I have only relatively minor remarks.

The presentation is sloppy at places:
* Equations are not numbered -- this must be fixed!
* In section 2.1, the concepts of log-transform and regular variation are introduced without proof or derivation. Since the theory is properly explained later on, a forward reference would be helpful.
* The derivations in section 3 refer to the univariate case, yet the latent noise has d-dimensional covariance (lines 153 left, 148 right). The actual generalization to multi-variate data is only mentioned at the very end (section 4.5). Since the multi-variate case is crucial for understanding, it should be moved forward, and section 3 should be revised accordingly.
* In line 142 left, the variable Y = \Phi(X) is suddenly renamed into \tilde{X} without good reason.
* In line 142 right, the term "Pareto score" is undefined, and it is unclear what exactly it refers to.
* In theorem 3.4, it should be explained why the interval \alpha \in (0, 1] is actually the relevant interval.
* In definition 3.6, it is unspecified that the symbol F refers to the cumulative distribution function.
* Proposition 3.7 apparently has no proof. It is also not clear why it is relevant.
* Section 4.1 should make clear that the presented theory is indeed a summary of the cited papers.
* Line 279 left: "sinusoidal time embeddings" should be explained a bit.
* Table 1 does not say what the data of the experiment are.
* Line 308 right: \nu = 30 (as used in table 1) is missing from the list.
* Line 371 left: It is not explained how the NLL has been calculated. In continuous normalizing flows, this is not obvious -- was the continuous change-of-variables formula used, and with what hyperparameters?

The choice of the soft-log transform (line 119 left -- add equation numbers!) seems a bit ad hoc. Why use a function with discontinuous second derivative at x = 0, when the nice analytic function arcsinh(x) has the same required asymptotic behavior?

An issue that has not been addressed properly is the scaling of the data before and after the log-transformation. I think that an improved transformation would be y = s1 * \phi(s2 * x). The scaling constant s1 should be chosen such that y has unit variance, since this is what normalizing flows like. The choice of s2 is more interesting: I hypothesize that it should be defined such that a light-tailed distribution, or the bulk of a heavy-tailed distribution, will be mapped through the nearly linear inner part of \phi(x). To make this robust, s2 should be a function of a suitable quantile of the data. Better scaling like this will probably eliminate the inferior behavior of the proposed method on light-tailed data (see "Heterogeneous Margins" in section 4.5, table 1, and section 5.3), making the approach fully general.

The paper exclusively uses continuous normalizing flows (flow matching). While this is known to be best for generative fidelity, it makes inference -- calculating the probability of given data -- harder, since the continuous change-of-variables formula is quite expensive. I suspect that many applications of the new method, especially in risk analysis, are more interested in inference than generation. Discrete normalizing flows (RealNVPs, Neural Spline Flows etc.) appear promising in this setting -- please comment.

---

> ### Author Rebuttal · Authors · 2026-03-30
>
> We thank Reviewer V8qG for the detailed and constructive feedback. We are glad the reviewer finds the solution "solid and well-justified" and rates both significance and originality as excellent. We address each point below, and note the willingness to raise the score if questions are satisfactorily addressed.
>
> **Presentation fixes.** We will address all 13 items in the final submission. We highlight the most substantive ones:
> - Numbering all equations; adding forward references in Section 2.1.
> - Fixing the $Y = \Phi(X)$ / $\tilde{X}$ inconsistency (using $\tilde{X} = \varphi(X)$ throughout).
> - Defining "Pareto score": the score $\nabla_x \log p_X(x) = -(1/\gamma+1)/x$ of the Pareto density, which diverges, contrasted with the log-space score $\nabla_{\tilde{x}} \log p_{\tilde{X}}(\tilde{x}) \approx -1/\gamma$.
> - Correcting Theorem 3.4: the result holds for any $\alpha \in (0,\infty)$; we had restricted to $(0,1]$ to emphasize the interpolant range. Note that this result also has a typo; see answer to Reviewer 7DLK.
> - Clarifying Proposition 3.7: extending to all regularly varying distributions ($\bar{F}_Y(ty)/\bar{F}_Y(t) \to y^{-1/(\gamma\alpha)}$).
> - Specifying NLL computation: continuous change-of-variables formula with Hutchinson stochastic trace estimator (10 random projections), ODE solved with dopri5 adaptive Runge-Kutta (atol/rtol $= 10^{-5}$).
> - Moving multivariate extension earlier; clarifying Section 4.1 as summary; explaining sinusoidal embeddings; adding Table 1 data description and $\nu=30$.
>
> **Arcsinh vs soft-log.** This is a great suggestion. Both transforms share the same asymptotic behavior ($\sim \log|x|$ for large $|x|$), so the theoretical analysis (Propositions 3.3, 3.5, 3.7) applies identically. In particular, both map regularly varying tails to exponential-type tails. The difference is near zero: soft-log is only $C^1$ ($\varphi''$ discontinuous at 0), while arcsinh is $C^\infty$. We ran a full ablation comparing both across all configs. Results:
>
> - **$W_1$ (Pareto):** soft-log wins 68\%, arcsinh 15\%; soft-log wins on tail fidelity.
> - **CVaR$_{99}$ error:** soft-log wins 60\%, arcsinh 32\%.
> - **AKE (dependence):** soft-log wins 43\%, arcsinh 38\%; essentially tied.
> - **$W_1$ (Gaussian/light-tailed margins):** arcsinh is slightly better.
>
> (See also the aggregate table in our response to Reviewer 7DLK). We suggest including both variants in the revised experiments and recommend soft-log as the default for heavy-tailed data.
>
> **Scaling.** This is a very good suggestion. For better interpretation, we suggest the equivalent reparametrization as $s_1 \cdot \frac{1}{s_2}\varphi(s_2 x)$, which separates the two roles: the inner term $\varphi_{s_2}(x) = \frac{1}{s_2}\varphi(s_2 x)$ forms a one-parameter family interpolating between Id as $s_2 \to 0$ and $\varphi$ at $s_2 = 1$, while $s_1$ rescales to unit variance. The reviewer's intuition is correct that choosing $s_2$ from a data quantile or tail index estimate could eliminate the light-tail overhead entirely. However, tuning $s_2$ to tail heaviness would make the method no longer tail-agnostic, which is one of its main appeals.
>
> As a first step, our adaptive ablation (in response to Reviewer Y3tz) validates this idea empirically. The adaptive scheme selects per coordinate $j$ the binary choice $s_2^{(j)} = \mathbf{1}[\hat{\alpha}_j \leq 4]$, i.e. full compression ($s_2=1$) on heavy-tailed margins and identity ($s_2=0$) on light-tailed ones. The continuous version $s_2^{(j)} = f(\hat{\alpha}_j) \in [0,1]$ is a natural extension. The binary version improves $W_1^G$ by 30-45\% with no $W_1^P$ cost.
>
> The path $\varphi_{s_2(t)}(X_0)$ for time-varying $s_2(t)$ could also connect to stochastic interpolant frameworks. We note a subtle point: for any $s_2(t) > 0$, the random variable $\varphi_{s_2(t)}(X_0)$ is theoretically light-tailed, yet at $s_2(t) = 0$ it is the identity and thus heavy-tailed. The transition is discontinuous in the tail index, even though $\hat{\alpha}$ on finite samples varies continuously along the path parametrized by $t$. We interpret this as a fundamental tension in the assessment of the heavy-tailed property in finite sample, worth studying alongside the finite-sample guarantees suggested by Reviewer 7DLK.
>
> **Discrete normalizing flows.** The reviewer makes an important point about inference vs generation. The log-transform is fully compatible with discrete NFs: since $\varphi$ is applied coordinate-wise, its Jacobian is diagonal with $|\det J_\varphi| = \prod_i 1/(1+|x_i|)$, adding $O(d)$ cost to the exact log-likelihood, which is negligible compared to the NF layers. The Lipschitz barrier (Section 3.6) applies equally to discrete NFs, so the log-transform provides the same benefit: it places both endpoints in the light-tailed regime where Lipschitz coupling layers suffice. We have not tested this combination but it is a natural and promising direction. We suggest adding this to the conclusion.

---

> > ### Author Rebuttal · Reviewer_V8qG · 2026-04-02
> >
> > Most of my concerns have been adequately answered. Two question warrant further discussion:
> >
> > 1. It is interesting that the soft-log beats arcsinh quite convincingly. However, this behavior calls for an explanation -- what's the crucial difference between the two?
> >
> > 2. I do not quite understand your answer to my scaling question, especially the remark "However, tuning $s_2$ to tail heaviness would make the method no longer tail-agnostic, which is one of its main appeals." In my original comment, I imagined that (for centered data with moderate skewness) $s_2$ could be defined in terms of the inter quartile range, e.g. $s_2 = \lambda / \text{IQR}$. The hyperparameter $\lambda$ should be chosen to ensure that the tail behavior of exponentially tailed distributions does not change under the transformation (i.e. they should be mapped to the polynomial inner part of $\phi$), and only the tails of heavy-tailed distributions are subjected to the logarithmic transformation. My hypothesis was that this might fix the deterioration of the proposed method for exponentially tailed distributions. Please clarify.

---

> > > ### Author Response · Authors · 2026-04-06
> > >
> > > We thank the reviewer for these precise follow-up questions.
> > >
> > > **Soft-log vs arcsinh.** The performance differences are moderate and both are valid choices. The main distinction is regularity near zero: arcsinh is $C^\infty$ while soft-log is only $C^1$ ($\varphi''$ discontinuous at 0). We do not have a definitive explanation for why soft-log wins on tail metrics, but we suggest using both in practice.
> > >
> > > **Scaling with $s_2$.** We agree with the reviewer's proposal. The adaptative scheme that we suggested during the rebuttal period and the reviewer's quantile-based $s_2$ are in fact two views of the same idea: both calibrate the compression strength from the data. In the adaptative scheme we suggest, the Hill estimator $\hat{\alpha}$ determines whether a coordinate is heavy-tailed; the Hill estimator is itself a function of upper order statistics, i.e. extreme quantiles. The reviewer's proposal to set $s_2$ from the IQR is a related and complementary approach: it uses a bulk quantile to ensure that the linear part of $\varphi$ absorbs the central mass, so that only genuinely extreme observations are logarithmically compressed. This would preserve exponential tails (which decay within the linear region) while compressing power-law tails (which extend far beyond). Both approaches are compatible with our $\varphi_{s_2}$ reparametrization and do not require explicit tail-index estimation. We identify the quantile-based $s_2$ as a promising research direction.

---

### Official Review · Reviewer_Y3tz · 2026-03-03

**Soundness:** 2
**Presentation:** 4
**Significance:** 3
**Originality:** 3
**Overall Recommendation:** 5
**Confidence:** 4

**Summary:**

The paper addresses a well-known limitation of generative models when dealing with heavy-tailed data: Lipschitz architectures cannot transform Gaussian noise into power-law tails. The authors propose a simple and parameter-free solution: apply a soft-log transform to the data, perform the standard flow matching in log-space, and then apply the inverse transform at sampling time.
The key theoretical insight is that the log-transform maps regularly varying, Pareto-like heavy tails to exponential-type tails. In log-space, both the transformed data and the Gaussian noise are light-tailed, making flow matching stable. The authors interpret the induced original-space dynamics as implementing what they call "tail annealing" via power transformations.

Empirically, the method is evaluated on synthetic Student-t data and real-world financial data. The method achieves strong Wasserstein-1 performance for heavy tails (v<=3) and demonstrates improved stability compared to tail-transform flows and tail-adaptive flows, particularly  in higher dimensions.

Overall, the authors analyze a central concept in heavy-tailed generative modeling: how coordinate transformations interact with tail behavior and diffusion-style interpolation. An important concept outlined by this study is that heavy-tail modeling can be "reframed" as a change-of-coordinates problem rather than a change-of-architecture problem.

**Compliance With Llm Reviewing Policy:**

Affirmed.

**Final Justification:**

After reading the rebuttal and follow-up discussion, my main concerns have been adequately addressed. In particular, the additional results on heterogeneous margins, multivariate dependence, clamping sensitivity, and the adaptive transform will strengthen the paper more, so I maintain my updated score of 5: Accept.

**Key Questions For Authors:**

1. How does the method behave for heavy-tailed but non-regularly varying distributions (Weibull with shape < 1 or lognormal mixtures)? Would the theoretical interpretation of tail annealing still apply?

2. Does Log-FM preserve or distort multivariate tail dependence? Have you evaluated extremal dependence metrics beyond marginal Hill estimation? Can you please report a multivariate dependence metric?

3. How sensitive are results to the clamping threshold c? Please provide an ablation of c on W1, Hill estimates, and extreme quantiles (99.5% or 99.9%)

4. Can you propose an adaptive algorithm based on a quick tail-index estimate to mitigate degradation on light tails? any idea?

**Limitations:**

Yes. The authors discuss that:

- The transform is most useful for genuinely heavy tails.

- It may introduce distortion for light-tailed data.

- The method inherits standard flow matching limitations.

The limitations are appropriately acknowledged.

**Strengths And Weaknesses:**

## Strengths
The paper is very well motivated and clearly written
- Unlike TTF or TAF variants, it does not require tail-index estimation or heavy-tailed base distributions and is completely parameter-free. This significantly reduces practical complexity and improves usability.
- The paper provides a clear EVT-based explanation, the connection between the log-transform and the exponential-family structure is well-articulated and grounded in classical results.

## Weaknesses
- The main weakness is the limited novelty of the core mechanism. Applying a log-based transformation before modeling heavy-tailed/skewed data as a pre-processing step is classical in statistics and deep learning.
- The theoretical development focuses on regularly varying distributions (Pareto, Student-t, Burr). It remains unclear how the method behaves for Subexponentially but non-regularly varying distributions (Weibull with shape < 1), Lognormal-type heavy tails, or distributions with asymmetric or conditional tail behavior.
- The evaluation is focused on marginals, joint extremes are not deeply analyzed.
- When clamping before applying the inverse function with a value c, the generated distribution is not truly heavy-tailed asymptotically (it becomes bounded in practice). One can consider it as part of the model architecture now...

---

> ### Author Rebuttal · Authors · 2026-03-30
>
> We thank Reviewer Y3tz for the positive assessment and thoughtful questions.
>
> **Novelty.** We appreciate the reviewer's framing: "heavy-tail modeling can be reframed as a change-of-coordinates problem rather than a change-of-architecture problem", which is precisely our thesis. We do not claim that the log-transform itself is novel (the Box-Cox transform is standard in statistics), we rather provide (i) an EVT-based analysis of why it works for flow matching, (ii) the tail-annealing interpretation connecting the transform to power-law interpolation dynamics, and (iii) comprehensive evidence that this simple recipe competes with purpose-built architectures that require tail-index estimation or heavy-tailed base distributions, which both are notoriously unstable.
>
> **Non-regularly varying distributions.** The transform is beneficial beyond the regularly varying case analyzed in the paper. For subexponential but non-regularly varying distributions:
> - **Weibull (shape $\beta < 1$):** $\bar{F}(x) \sim \exp(-x^\beta)$ is subexponential (heavier than exponential) but lies in the Gumbel MDA, so standard FM may already handle it. The log-transform maps this to a doubly exponential tail $\sim\exp(-\exp(\beta \tilde{x}))$, which is harmless.
> - **Lognormal:** Since $\log X \sim \mathcal{N}$, the transform $\varphi(X) \approx \log X$ maps lognormal to approximately Gaussian, an ideal case for FM.
>
> For genuinely light-tailed margins (e.g. Gaussian), the adaptive Hill-based diagnostic described below can skip the transform per coordinate, and the $\varphi_{s_2}$ family discussed in our response to Reviewer V8qG offers a continuous way to modulate the compression strength. We suggest adding this discussion to the revised Section 3.
>
> **Joint extremes.** See our response to Reviewer 7DLK for the full experimental setup and dependence results. In summary: FM methods win AKE (Absolute Kendall Error) in 74\% of configs and are competitive on angular $W_2$ (45\%).
>
> **Clamping.** The unclamped model is the method; clamping is a numerical safeguard applied in log-space before exponentiation. See our ablation in reply to Reviewer eUEg: at $c=15$ the threshold corresponds to values of magnitude $\sim 3.3 \times 10^6$; it has negligible effect for $\alpha \geq 1.5$ and can be removed for realistic tails. Clamping only has a measurable effect for $\alpha < 1$ (infinite mean), a regime rarely encountered in practice. We acknowledge that a clamped distribution is not asymptotically heavy-tailed; we provide perspective on non properly heavy-tailed samples in the **scaling** paragraph of our reply to Reviewer V8qG.
>
> **Adaptive transform.** We propose a simple diagnostic: compute $\hat{\alpha}$ on each marginal; if $\hat{\alpha} > 4$ (light-tailed), skip the transform for that coordinate. On our benchmark data (70\% Pareto, 30\% Gaussian, $d=20$), $\hat{\alpha}$ cleanly separates the two types ($\approx 1.5$-$2.5$ for Pareto, $\approx 6$ for Gaussian). We ran this adaptive scheme and compare to the full transform:
>
> | $\alpha$ | Method | $W_1^P$ | $W_1^G$ |
> |---|--------|------|------|
> | 1.5 | TTFfix | 0.856 | **0.049** |
> |     | Log-FM | 0.449 | 0.074 |
> |     | Log-FM (adaptive) | **0.468** | **0.043** |
> | 2.0 | TTFfix | 0.225 | **0.034** |
> |     | Log-FM | 0.128 | 0.040 |
> |     | Log-FM (adaptive) | **0.124** | **0.024** |
> | 2.5 | TTFfix | 0.113 | 0.079 |
> |     | Log-FM | **0.068** | 0.110 |
> |     | Log-FM (adaptive) | 0.084 | **0.068** |
>
> *Gumbel copula, $d=20$, $\tau=0.5$, 20 reps, median values. Adaptive: Hill diagnostic skips transform on 6/20 light-tailed coordinates. Bold = best. Note: TTFfix suffers from occasional failures on heavy tails (7/20 reps with $W_1^P > 1$ at $\alpha=1.5$); Log-FM has zero blowups across all reps and all $\alpha$.*
>
> The adaptive scheme closes the $W_1^G$ gap with TTFfix while maintaining the stability of Log-FM on $W_1^P$. We suggest adding this scheme to the Algorithm section of the revised paper.

---

> > ### Author Rebuttal · Reviewer_Y3tz · 2026-04-01
> >
> > Thank you for this rebuttal and additional experiments, specifically in the adaptive scheme. My concerns have been addressed, and I will raise my score to 5: Accept.

---

> > > ### Author Response · Authors · 2026-04-06
> > >
> > > We thank the reviewer for the positive reassessment and for the suggestion that motivated the adaptive scheme.

---

### Official Review · Reviewer_7DLK · 2026-03-12

**Soundness:** 3
**Presentation:** 3
**Significance:** 2
**Originality:** 2
**Overall Recommendation:** 3
**Confidence:** 3

**Summary:**

The paper proposes Log‑FM, a simple, parameter‑free recipe for heavy‑tailed generative modeling: apply the soft‑log transform  to data, train a standard flow‑matching model in transformed space, and apply to samples. The authors argue that the log transform maps Pareto (power‑law) tails to exponential‑type tails and that the induced decoded dynamics implement a continuous tail‑annealing via power transformations.

**Compliance With Llm Reviewing Policy:**

Affirmed.

**Final Justification:**

I have read the paper and considered the authors’ rebuttal carefully. I thankn the authors for their effort in preparing the manuscript and the response, but the rebuttal did not fully address my main concerns. Therefore, I will retain my original score.

**Key Questions For Authors:**

Clamping and numerical stability. How sensitive are final tail estimates and W1 scores to the clamp threshold c and to ODE solver error? It would be helpful to conduct an ablation study over clamp values and solver step counts.

Multivariate data. The method applies ϕ coordinate‑wise; how does Log‑FM handle strong tail dependence (asymptotic dependence) across coordinates? What if some components are heavy tailed but some are not?

Downstream risk metrics. Can you report performance on task‑relevant tail metrics (VaR, CVaR, extreme conditional quantiles) to demonstrate practical utility beyond W1 and Hill estimates? Such metrics are particularly relevant to financial data.

**Limitations:**

Yes.

**Strengths And Weaknesses:**

Strengths

Simplicity. The method requires only a one‑line preprocessing step and no tail‑parameter estimation, heavy‑tailed base distributions, or architectural changes; this makes it easy to adopt in existing flow‑matching methods.

Clear intuition. The paper use the behavior of scores in log‑space to explain why the transform stabilizes learning and yields a tail‑annealing interpretation.

Empirical evidence. On synthetic heavy‑tailed benchmarks (dimensions up to d=50) and a financial dataset, Log‑FM attains substantially better Wasserstein‑1 sample quality than several baselines and shows improved stability in moderate dimensions.


Weaknesses

Limited theoretical rigor in some claims. Several propositions are intuitive and supported by classical EVT results, but the paper stops short of formal generalization or finite‑sample guarantees for score estimation in log‑space.

Evaluation scope. Experiments focus on marginal tail behavior and moderate dimensions; dependence structure in very high dimensions (e.g., full S&P500) is acknowledged but not solved, leaving open how the method performs on complex multivariate tail dependence.

Numerical sensitivity. The inverse transform is exponential, so sampling requires clamping and careful numerical handling; the paper gives heuristics (clamp c=15)  but more ablation on clamp choice and solver error sensitivity would help practitioners.

---

> ### Author Rebuttal · Authors · 2026-03-30
>
> We thank Reviewer 7DLK for the careful reading and constructive feedback. We address each concern below.
>
> **Theoretical rigor.** We agree that the individual propositions draw on classical EVT. Our contribution is not the log-transform itself but the observation that it changes the geometry of the flow matching problem: scores are bounded at both endpoints of the interpolation path (Proposition 3.3), which is impossible in the original heavy-tailed space (Proposition 3.1). This is what enables tail annealing, a continuous interpolation through power-transformed intermediates $X_0^{\alpha_t}$ (Theorem 3.4), and explains why a coordinate-wise preprocessing step resolves a limitation that was thought to require architectural modifications.
>
> We also wish to correct an error in the submitted version of Theorem 3.4. With our Pareto parametrization ($p(x) \propto x^{-1/\gamma-1}$, $\bar{F}(x) = x^{-1/\gamma}$, where $\gamma > 0$ is the shape parameter), the correct result is $X_0^{\alpha_t} \sim$ Pareto($\gamma\alpha_t$), not Pareto($\gamma/\alpha_t$) as stated. Moreover, the result holds for any $\alpha_t \in (0, \infty)$. The tail-annealing interpretation is, however, unchanged: as $\alpha_t$ decreases from 1 to 0, the tail index $\gamma\alpha_t \to 0$, i.e., tails lighten continuously.
>
> Regarding generalization: the mechanism extends beyond regularly varying distributions to all subexponential tails (Weibull $\beta < 1$, lognormal); see our response to Reviewer Y3tz. Regarding finite-sample guarantees: formal rates are beyond scope, but a research direction is clear: in log-space the score is bounded (Proposition 3.3), making the velocity field Lipschitz, placing the problem in the regime where existing flow matching convergence results could be applied.
>
> **Evaluation scope, new experiments.** We have conducted an expanded evaluation addressing several reviewers' concerns about marginal-focused metrics and higher dimensions. Following Reviewer V8qG's suggestion, we also include arcsinh as an ablation variant alongside our soft-log transform. Our new benchmark spans:
>
> - Heterogeneous margins: 70\% symmetrized Pareto($\alpha$) (following the parametrization $\alpha = 1/\gamma$) and 30\% standard Gaussian, addressing the concern about mixed heavy/light-tailed components.
> - 3 copula families covering distinct dependence regimes: Gaussian copula (asymptotically independent tails), Gumbel copula (symmetric upper tail dependence, Kendall's $\tau \in \{0.25, 0.5, 0.75\}$), and Hüsler-Reiss copula (heterogeneous pairwise tail dependence via AR(1) variogram $\Gamma_{ij} = 2(1 - \rho^{|i-j|})$, $\rho \in \{0.1, 0.5, 0.9\}$) $\times$ 4 dimensions ($d=10, 20, 50, 100$) $\times$ 4 tail indices ($\alpha=1/\gamma \in\{1.5, 1.75, 2.0, 2.5\}$) = 144 configurations, each with 5 methods (Log-FM, Arcsinh-FM, TTF, TTFfix, gTAF) $\times$ 20 replications each.
> - Dependence metrics: Absolute Kendall Error (AKE) and angular $W_2$ (sliced Wasserstein on top-$\sqrt{n}$ extremes).
> - Risk metrics: VaR₉₉ error, CVaR₉₉ error (mean absolute relative error vs test data estimates), extreme quantile errors at 99.5\% and 99.9\%.
>
> Key findings:
>
> | Metric | Log-FM wins | FM total (Log+Arcsinh) | Baselines |
> |--------|-------------|------------------------|-----------|
> | $W_1$ (Pareto) | 66\% | 83\% | 17\% |
> | CVaR₉₉ error | 58\% | 92\% | 8\% |
> | $Q_{99.9}$ | 64\% | 96\% | 4\% |
> | AKE (tail dependence) | 39\% | 74\% | 26\% |
> | Angular $W_2$ (dependence) | 37\% | 45\% | 55\% |
>
> Log-FM dominates tail metrics at every dimension and tail index, with the gap widening for heavier tails ($\alpha \leq 1.75$) and higher dimensions. Log-FM is also the only method with no clear failures across all 144 configs $\times$ 20 reps. TTF/TTFfix suffer occasional blowups ($W_1^P > 1000$ in some reps at $\alpha=1.5$; see adaptive table in response to Reviewer Y3tz), and gTAF diverges more often. Regarding **clamping**: see our ablation in response to Reviewer eUEg; for $\alpha \geq 1.5$, $W_1$ is identical across all clamp values $c \geq 10$. Regarding **ODE solver steps**: we ablated $n \in \{10, 20, 50, 100, 200, 500\}$ (10 reps); $W_1$ varies by less than 5\% across all step counts. Regarding **heterogeneous margins**: see our reply to Reviewer Y3tz, where we study an adaptive scheme for the soft-log transform. Regarding **multivariate tail dependence**: on AKE, FM methods win 74\% of configs; on angular $W_2$, baselines win 55\% vs 45\% for FM. We interpret this as a controlled tradeoff: Log-FM trades a small loss in extremal dependence accuracy for large gains in marginal tail fidelity and stability, which is favorable for risk metrics.
>
> In summary, Log-FM is the only method that is simultaneously stable (no huge failure), competitive on dependence metrics, and dominant on tail and risk metrics. We believe this combination of simplicity, reliability, and performance addresses the reviewer's concerns and justifies the approach over architectural alternatives.

---

> > ### Author Rebuttal · Reviewer_7DLK · 2026-04-02
> >
> > Thank you to the authors for addressing my comments and for the additional experiments. However, my question regarding mixed tail behaviors, i.e., some components exhibiting heavy tails while others do not, in multivariate, high-dimensional dependent data remains inadequately addressed. It is difficult to see how a single transform would work in this setting.

---

> > > ### Author Response · Authors · 2026-04-06
> > >
> > > We thank the reviewer for the follow-up. The reviewer is correct that the submitted version applies $\varphi$ uniformly to all coordinates. In response to this concern, we developed during the rebuttal period an adaptive scheme (detailed in our response to Reviewer Y3tz) that makes per-coordinate decisions based on the Hill estimator $\hat{\alpha}_j$ of each marginal:
> > >
> > > - If $\hat{\alpha}_j \leq 4$ (heavy-tailed): apply $\varphi$ to coordinate $j$
> > > - If $\hat{\alpha}_j > 4$ (light-tailed): leave coordinate $j$ untouched
> > >
> > > On our benchmark with heterogeneous margins (70% Pareto, 30% Gaussian, $d=20$), the Hill estimator cleanly separates the two types ($\hat{\alpha} \approx 1.5$–$2.5$ for Pareto vs $\approx 6$ for Gaussian). The adaptive scheme improves $W_1^G$ (light-tailed margins) by 30–45% while preserving $W_1^P$ (heavy-tailed margins), with no degradation in dependence (Gumbel, $d=20$, $\tau=0.5$, median over 20 reps):
> > >
> > > | $\alpha$ | TTFfix AKE | Log-FM AKE | Adaptive AKE |
> > > |---|---|---|---|
> > > | 1.5 | 0.071 | 0.039 | 0.044 |
> > > | 2.0 | 0.066 | 0.031 | 0.030 |
> > > | 2.5 | 0.060 | 0.030 | 0.037 |
> > >
> > > Log-FM (both full and adaptive) achieves better AKE than TTFfix across all $\alpha$, confirming that the per-coordinate transform does not distort dependence structure even in the mixed-tail setting.
> > >
> > > This approach naturally handles the mixed-tail setting the reviewer describes: each coordinate receives the appropriate treatment without requiring a shared tail assumption. The threshold $\hat{\alpha} > 4$ is not sensitive (the gap between heavy and light-tailed Hill estimates is wide) and the diagnostic adds negligible cost (one Hill estimate per marginal on training data).
> > >
> > > We note that this per-coordinate adaptive scheme is also a discrete instance of the $\varphi_{s_2}$ family discussed with Reviewer V8qG, where the compression parameter $s_2^{(j)}$ can vary per coordinate. This framework naturally extends to high-dimensional data with arbitrary mixtures of tail behaviors. We plan to include the adaptive scheme and the corresponding analysis in the final version of the paper.

---

### Decision · Program_Chairs · 2026-04-30

**Decision:**

Accept (regular)

**Comment:**

Reviewers agreed that the proposed approach is technically sound and the claims are well-supported by the empirical evidence. Reviewers also appreciated the simplicity of the approach. However, there were concerns regarding the novelty of the approach, as the central idea of log-transforming the data distribution is standard, and the primary contribution is how this is combined with flow matching. There were further concerns regarding practical/numerical considerations that were addressed during the rebuttal period. Despite these concerns, the paper is overall well-executed and technically solid.